# Loss of microglial MCT4 leads to defective synaptic pruning and anxiety-like behavior in mice

Katia Monsorno [1], Kyllian Ginggen[1], Andranik Ivanov[2], An Buckinx[1], Arnaud L. Lalive[3], Anna Tchenio[3], Sam Benson[4], Marc Vendrell [4], Angelo D'Alessandro[5], Dieter Beule[2], Luc Pellerin [6], Manuel Mameli [3] & Rosa Chiara Paolicelli [1] ✉

Microglia, the innate immune cells of the central nervous system, actively participate in brain development by supporting neuronal maturation and refining synaptic connections. These cells are emerging as highly metabolically flexible, able to oxidize different energetic substrates to meet their energy demand. Lactate is particularly abundant in the brain, but whether microglia use it as a metabolic fuel has been poorly explored. Here we show that microglia can import lactate, and this is coupled with increased lysosomal acidification. In vitro, loss of the monocarboxylate transporter MCT4 in microglia prevents lactate-induced lysosomal modulation and leads to defective cargo degradation. Microglial depletion of MCT4 in vivo leads to impaired synaptic pruning, associated with increased excitation in hippocampal neurons, enhanced AMPA/GABA ratio, vulnerability to seizures and anxiety-like phenotype. Overall, these findings show that selective disruption of the MCT4 transporter in microglia is sufficient to alter synapse refinement and to induce defects in mouse brain development and adult behavior.

Microglia, the resident macrophages of the central nervous system (CNS), are implicated in a variety of biological processes and critically contribute to proper brain development and brain homeostasis across the lifespan[1–3]. They are highly motile and constantly scan the brain parenchyma making contact with neighboring cells[4,5]. Microglia sample the surrounding microenvironment by rapidly extending and retracting their processes, ensuring continuous nanoscale surveillance through actin-dependent filopodia motility[6]. Such a dynamic activity requires ceaseless ATP and cAMP-dependent cytoskeleton rearrangements[7,8]. To adequately meet the high energy demand for sustaining a constant surveying activity, microglia must display metabolic flexibility and be able to utilize a variety of metabolic substrates according to their availability[9]. While oxidative metabolism of glucose represents a major source of energy under homeostatic condition, other substrates have been shown to be efficiently oxidized by microglia[10]. Consistent with a rapid metabolic adaptation, microglia can maintain their scanning activity and preserve tissue surveillance even in the absence of glucose, by using alternative sources like glutamine or fatty acids as an energetic fuel[11]. Whether other metabolites, such as lactate, could serve as similar energetic substrates for microglia, thus contributing to brain homeostasis, it remains poorly explored.

Lactate is a metabolic product of glycolysis, which results from the reduction of pyruvate. Once released extracellularly, lactate can

[1]University of Lausanne, Department of Biomedical Sciences, Lausanne, Switzerland. [2]Core Unit Bioinformatics, Berlin Institute of Health, Charité—Universitätsmedizin Berlin, Berlin, Germany. [3]University of Lausanne, Department of Fundamental Neurosciences, Lausanne, Switzerland. [4]University of Edinburgh, Centre for Inflammation Research, Edinburgh, United Kingdom. [5]University of Colorado, Anschutz Medical Campus, Department of Biochemistry and Molecular Genetics, Denver, CO, USA. [6]Inserm U1313, University of Poitiers and CHU of Poitiers, Poitiers Cedex, France. ✉e-mail: rosachiara.paolicelli@unil.ch

act as a signaling molecule, or be imported and utilized by other cells as a bioenergetic substrate[12,13]. In the brain, lactate acts as a key molecule to sustain and regulate neuronal activity[14–16]. Highly glycolytic astrocytes and oligodendrocytes can directly shuttle lactate to neurons, to support neuronal oxidative phosphorylation via lactate dehydrogenase (LDH), the enzyme that catalyzes the reversible conversion of pyruvate and NADH to lactate and NAD+ in a dynamic equilibrium[17]. Of note, this enzyme is found either as a homotetramer or a heterotetramer of two different subunits (LDHA or M subunit, encoded by the *LDHA* gene, and LDH-B or H subunit, encoded by the *LDHB* gene). LDH complex containing four B subunits (LDH-1) favors catabolizing lactate into pyruvate, whereas the opposite is true for the isoform containing four A subunits (LDH-5)[14,18,19]. Lactate shuttles operate also in other tissues and cell types, both in physiological and pathological conditions. For instance, cancer cells have been largely described to act in metabolic symbiosis, with lactate-producing cells being in close metabolic cooperation with lactate-consuming cells, thus confirming that lactate is not merely a waste product of glycolysis, but rather a key metabolite for inter-cellular communication[20,21]. Similarly, heterotypic cell metabolic cross-talks between endothelial cells and residential macrophages have been reported to involve lactate in the context of inflammatory diseases[22,23]. For lactate to be shuttled between cells, monocarboxylate transporters (MCTs) are required, which allow the simultaneous transfer of one lactate molecule and one proton, following the concentration gradient[24,25].

Here, we found that microglia can efficiently import lactate, and this promotes lysosomal acidification. Among the major lactate transporters expressed in the brain, only MCT4 appears to be dynamically regulated in microglia upon exposure to exogenous lactate. We show that the lactate-dependent lysosomal modulation in microglia requires MCT4, and that cells lacking this transporter are deficient in cargoes uptake and degradation. During early postnatal development, conditional knockout mice (cKO) selectively lacking MCT4 in microglia, display impaired synaptic pruning, associated with increased levels of synaptic markers in the CA1 hippocampal region. At the functional level, hippocampal pyramidal neurons of cKO mice receive increased excitatory drive, as supported by larger amplitude of excitatory post-synaptic currents and display increased excitatory to inhibitory (AMPA/GABA) ratio. Furthermore, juvenile mice lacking microglial MCT4 exhibit susceptibility to kainic acid-induced seizures, suggestive of circuit hyperexcitability. At the behavioral level, adult MCT4 cKO mice display anxiety-like phenotypes. Altogether, these findings show that selective disruption of the MCT4 transporter in microglia is sufficient to alter microglia-mediated synapse refinement and to induce defects in brain development and adult behavior.

## Results

### Microglia are metabolically versatile cells and can efficiently import lactate

To assess the transcriptional bases of metabolic flexibility, we analyzed the expression profile of microglia acutely isolated from the brain of adult Cx3cr1GFP mice (dataset from Pinto et al.[26], ArrayExpress database E-MEXP-3347) in relation to the major intracellular KEGG metabolic pathways. We found that most of the genes within each metabolic pathway were highly expressed in microglia (Fig. 1a, Supplementary Fig. 1a). We also analyzed the expression KEGG pathway genes through module scores in microglia as compared to other brain cell populations (neurons, astrocytes, and oligodendrocytes), acutely isolated from the adult mouse hippocampus (scRNA-seq dataset from Mattei et al.[27]). We found that, in particular, microglia display a relative high module score for oxidative phosphorylation and glycolysis as compared to other cell types (Fig. 1b, Supplementary Fig. 1b). Altogether, these data are indicative of an intrinsic ability of microglia, in principle, to metabolize a large variety of energetic substrates.

*LDHB* appeared as one of the most abundant genes implicated in microglial metabolic regulation (Fig. 1a). Furthermore, *LDHB* is highly enriched in microglia among all the other brain cell types in the adult mouse hippocampus[9,28] and its expression peaks in these cells in the second postnatal week[29], a time of high synapse remodeling. Since LDHB-containing isoforms are enriched in cells preferentially converting lactate into pyruvate, we hypothesized that its high levels in microglia may underlie a prominent role for lactate metabolism.

First, we asked whether microglia could import exogenously administered lactate. We took advantage of fluorescent analogs of metabolites (SCOTfluors) and used fluorescent lactate[30] to assess lactate import in primary microglia by live confocal imaging. The fluorescent SCOTfluor lactate was detected inside microglia already after one hour of incubation, and it was significantly higher when glucose availability in the culture medium was reduced (25 mM vs. 0.1 mM glucose; Fig. 1c, d). Furthermore, we confirmed by mass spectrometry that primary microglia exposed to 20 mM sodium lactate display an increase of about fourfold in lactate levels compared to control, pointing towards a significant import of lactate in these cells (Fig. 1e). Overall, these findings support that microglia are able to internalize extracellular lactate.

### The lactate transporter MCT4 is upregulated in response to extracellular lactate and is required for lactate-induced lysosomal acidification

Lactate can be transported across cell membranes through a class of proton-linked solute carrier members (SLC16), that carry molecules with one carboxylate group (monocarboxylates), such as lactate, pyruvate, and ketones, therefore defined as monocarboxylates transporters (MCTs)[24,25]. Of the 14 isoforms comprising the MCTs family, the lactate transporters highly abundant in the brain are MCT1, MCT2, and MCT4, which are also expressed by microglia[31]. Thus, we asked whether one of those could be the main responsible for lactate transport in these cells. Upon exposure to 25 mM sodium lactate for 24 hours, we observed a significant upregulation only in the Slc16a3 transcript, encoding for MCT4 (Fig. 2a) in primary microglia, while no changes were reported for MCT1 or MCT2 (Fig. 2a). Lactate exposure also significantly increased MCT4 at the protein level (Fig. 2b, c). These findings support a direct implication of MCT4 in the microglial response to extracellular lactate availability.

Next, we aimed to further elucidating the downstream regulation exerted by MCT4. In oxidative cells, lactate import has been shown to mediate lysosomal acidification and proteolytic function through LDHB activity[32]. Therefore, given the high expression of LDHB in microglia, and their ability to internalize lactate, we hypothesized that a similar intracellular modulation would also occur in microglia through MCT4.

In order to assess whether this is the case, we generated an inducible microglia-specific conditional MCT4 knockout mouse line, by crossing Cx3cr1^CREert2 and Slc16a3^flox mice[33,34], to produce Cx3cr1^CREert2/+;Slc16a3^flox/flox (MCT4 cKO) and Cx3cr1^CREert2/+;Slc16a3^+/+ (control) littermates (Fig. 2d). These mice were used to prepare primary microglial cultures and MCT4 KO was induced in vitro by 4-hydroxytamoxifen treatment. The efficiency of MCT4 depletion was confirmed by Western blot, showing more than 90% reduction as compared to controls (Fig. 2e, f).

Primary microglia cultures were maintained in astrocyte-conditioned medium, which contained between 5 mM and 7 mM lactate, as measured spectrophotometrically[35] (Supplementary Fig. 2a). Confocal imaging of the pH-dependent dye LysoTracker revealed that MCT4 KO microglia displayed significantly less area covered by acidic organelles (late endosomes and lysosomes) (Fig. 2g, h), suggesting that they could not possibly utilize the lactate available in the basal culture medium as efficiently as control cells. Conversely, no differences in LysoTracker signal were

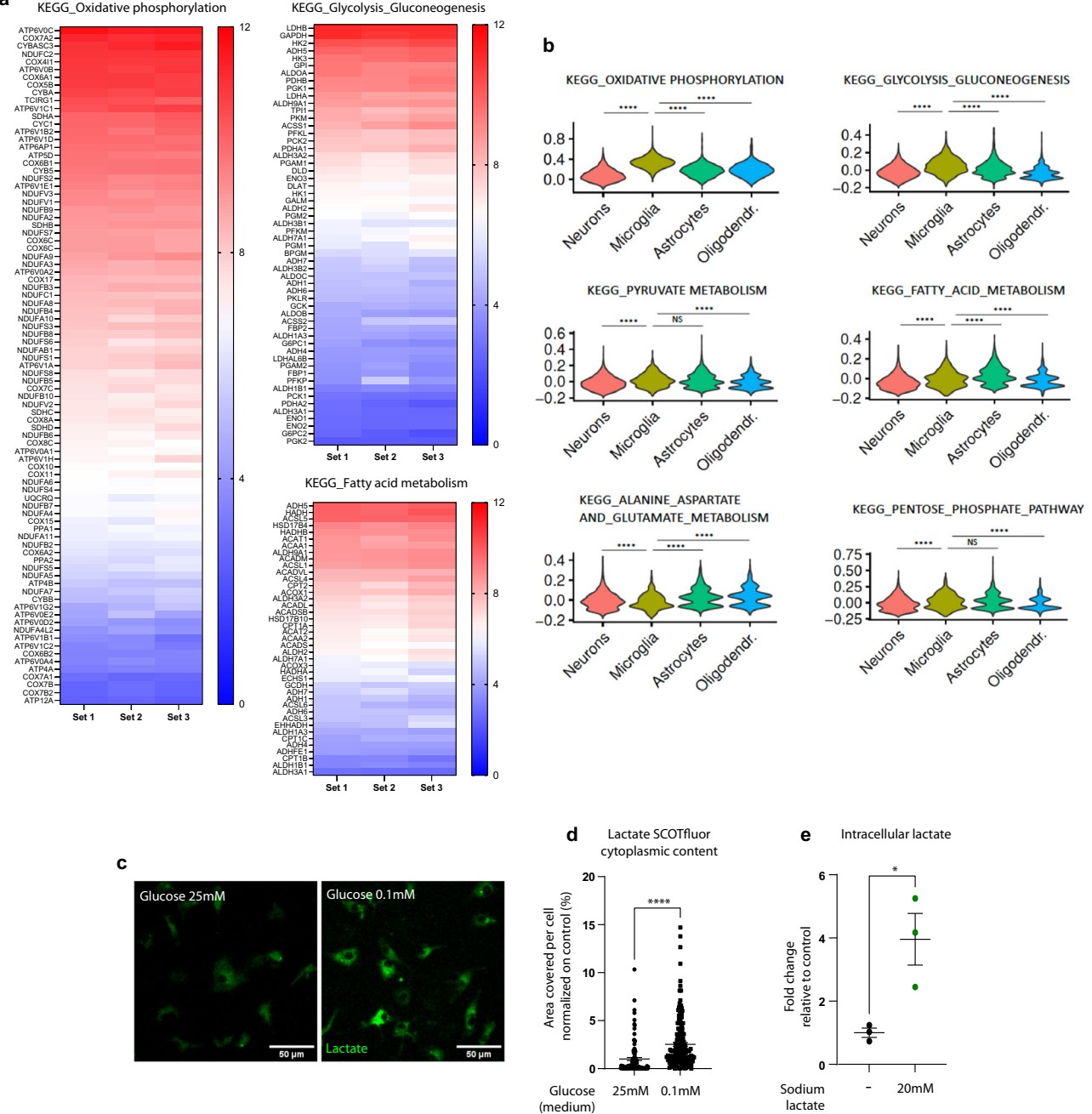

**Fig. 1 | Microglia are metabolically flexible and can efficiently import lactate in vitro. a** Heatmaps of the microglial expression levels of genes annotated within substrate metabolism-related KEGG pathways. Data were obtained from the dataset published in Pinto et al.[26], from three independent experiments, shown as normalized intensity values. **b** Violin plot showing Seurat[83] module scores for six KEGG pathways across different cell types in the mechanically dissociated mouse hippocampus (dataset from Mattei et al.[27]). The module scores represent the average expression of all genes in a particular cluster at a single-level, subtracted by the aggregated expression of control feature sets. Module scores in microglia are compared to other cell types; Two-sided Wilcoxon test, ****$p < 0.0001$; exact

$p$ values are provided in Supplementary Fig. 1b. **c** Representative confocal z-stack projections of SCOTfluor lactate internalized by primary microglia, in standard (25 mM) or reduced (0.1 mM) glucose conditions and **d** relative quantification. Data point represent individual cells; glucose 25 mM: $n = 105$ cells; glucose 0.1 mM: $n = 168$ cell from $N = 3$ independent experiments. Two-tailed unpaired $t$ test, ****$p < 0.0001$. Scale bar: 50 μm. **e** Intracellular lactate content measured by LC-MS in primary microglia cells upon 6 h exposure to 20 mM Na-lactate or control in standard 25 mM glucose conditions. $N = 3$ independent experiments. Two-tailed unpaired $t$ test, *$p = 0.0235$. **d**, **e** Data are represented as mean ± SEM. **a**, **b**, **d**, **e** Source data are provided as a Source Data file.

observed when MCT4 KO and control microglia were cultured in lactate-free medium (Supplementary Fig. 2b). To further test whether lactate was directly implicated in lysosomal acidification, we exposed primary microglia to 25 mM lactate for 24 hours. While no morphological changes were observed upon treatment (Supplementary Fig. 2c), we found an increase of about 50% in the area covered by acidic organelles in control cells as compared to vehicle

controls (Fig. 2i, j). Of note, lactate administration completely failed to induce any modulation in the absence of MCT4 (Fig. 2i, j). Independent experiments confirmed that lactate exposure leads to enhanced LysoTracker signal, without altering the levels of the lysosomal-associated membrane protein LAMP1, thus supporting a specific effect on lysosomal acidification, independent of structural changes (Supplementary Fig. 3a–d).

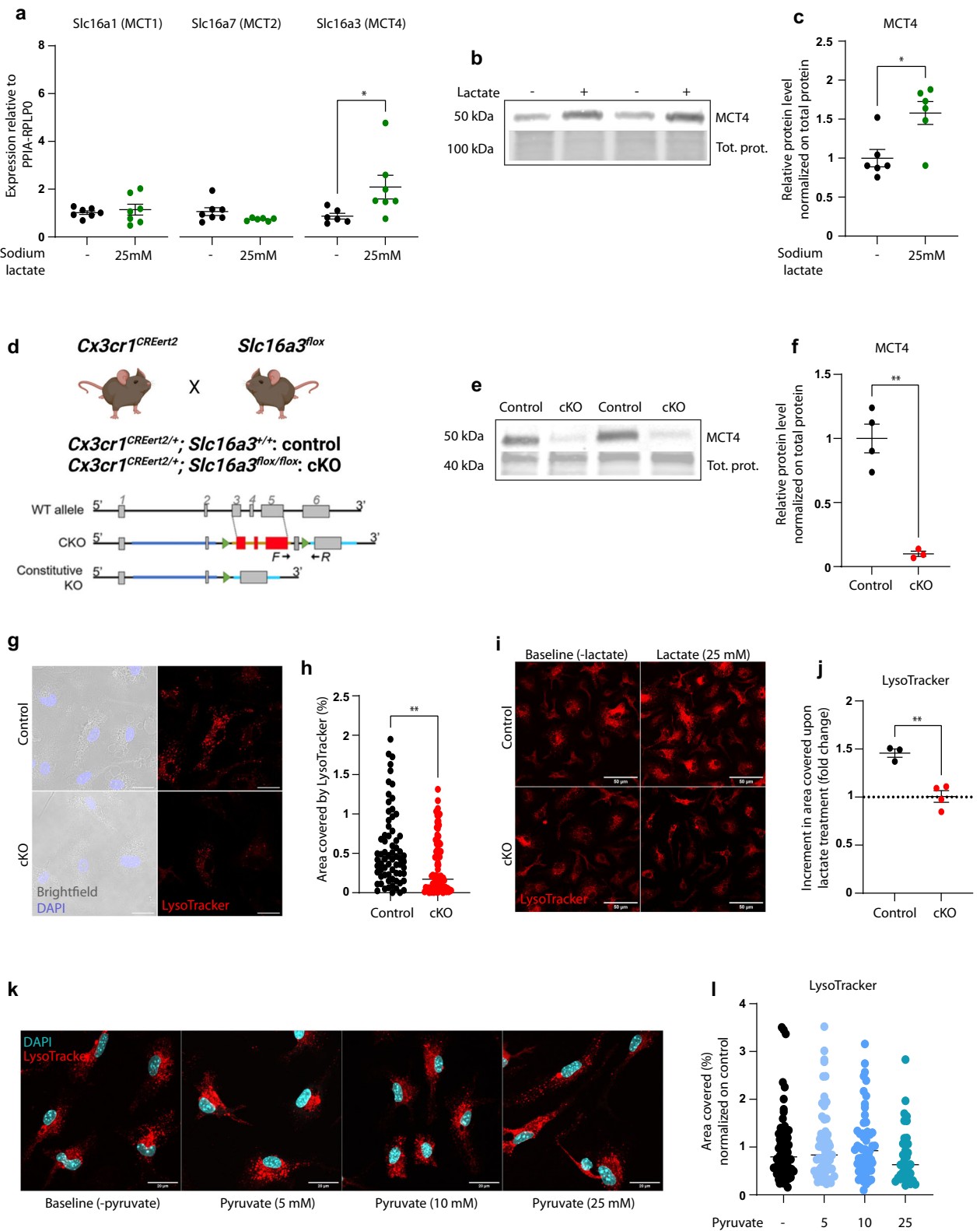

We exposed primary microglia to increasing concentrations of lactate and found that, while exposure to 1 mM lactate was not sufficient to significantly increase the LysoTracker signal in microglia, 3 mM and 5 mM elicited a response similar to 25 mM lactate (Supplementary Fig. 3e, f). On the contrary, primary microglia exposed to similar concentrations of pyruvate did not show any modulation of the area covered by LysoTracker (Fig. 2k, l), confirming that the observed

effect on lysosomal acidification is specifically induced by lactate and not by other monocarboxylates.

Lactate catabolism by LDHB-enriched isozymes has been proposed to promote lysosomal acidification through v-ATPase, a proton pump of lysosomes[32]. The V-ATPase inhibitor bafilomycin A1 completely prevented the increase in LysoTracker signal induced by lactate exposure in microglia (Supplementary Fig. 3g, h), indicating that the

**Fig. 2 | The lactate transporter MCT4 is upregulated in microglia in response to extracellular lactate and is required for lactate-dependent lysosomal acidification. a** mRNA expression of the lactate transporters Slc16a1, Slc16a7, and Slc16a3 in primary microglia in the presence or absence of 25 mM lactate (24 h treatment). Each data point represents the average of $n = 3$ technical replicates from $N = 7$ independent experiments. Two-tailed unpaired $t$ test, *$p$(Slc16a3) = 0.0481. **b** Representative western blot of MCT4 protein expression in primary microglia in standard medium (−) or medium containing 25 mM lactate (+, 24 h treatment), and **c** relative quantification from $N = 6$ independent experiments. Two-tailed unpaired $t$ test, *$p = 0.0104$. **d** Schematic of the breeding strategy for producing Cx3cr1CRE^{ERT2};MCT4^{flox} mice and representation of the floxed exons in the MCT4 gene (Slc16a3). Created with BioRender.com. **e** Representative western blot for MCT4 depletion upon 4-hydroxytamoxifen treatment in primary microglia from control and cKO mice and **f** relative quantification, normalized on total protein. Data points represent primary microglia prepared from one individual, from $N = 4$ control and $N = 3$ cKO. Two-tailed unpaired $t$ test, **$p = 0.0011$. **g** Representative confocal z-stack projections of LysoTracker labeling in control and cKO primary microglia and **h** relative quantification of the area covered by LysoTracker signal per cell. Data points represent individual cells, control: $n = 79$ cells; cKO: $n = 84$ cells from $N = 2$ independent experiments. Two-tailed unpaired $t$ test, **$p = 0.0016$. Scale bar: 20 μm. **i** Representative confocal z-stack projections of LysoTracker staining in control and cKO primary microglia, at baseline and after 25 mM lactate exposure, and **j** signal quantification, displayed as fold change compared to baseline. Data point represent primary microglia prepared from single individuals (control $N = 3$; cKO $N = 4$), from three independent experiments. Two-tailed unpaired $t$ test. **$p = 0.0025$. Scale bar: 50 μm. **k** Representative confocal z-stack projections of LysoTracker labeling in primary microglia exposed to increasing concentration of pyruvate and **l** relative quantification of the area covered by LysoTracker signal per cell, normalized to control. Data points represent individual cells from $N = 2$–3 independent experiments. Baseline (-pyruvate): $n = 64$ cells, Pyruvate (5 mM): $n = 57$ cells; Pyruvate (10 mM): $n = 57$ cells; Pyruvate (25 mM): $n = 35$ cells. **a**, **c**, **f**, **h**, **j**, **l** Data are represented as mean ± SEM. Source data are provided as a Source Data file.

lactate-induced modulation of lysosomal acidification depends on V-ATPase activity. Overall, these findings indicate that extracellular lactate leads to enhanced acidification of late endosomes and lysosomes in microglia, and that this is strictly dependent on the MCT4 transporter. Microglia lacking MCT4 display defective lactate-induced acidification of degradative organelles, suggesting that their phagocytic function might be compromised.

## Selective depletion of MCT4 in microglia impairs phagocytosis

Low intraluminal pH of lysosomes is critical for optimal degradation of substrate cargo[36]. Therefore, we asked whether microglial MCT4 depletion could lead to dysfunction in the microglial phagocytic capacity. To this aim, we performed DQ-BSA assay in both control and KO primary cells. DQ-BSA is a fluorogenic substrate for proteases, whose fluorescence is quenched by its labeling with BODIPY dyes. Upon hydrolysis of the DQ-BSA to single peptides by proteases, the quenching is relieved, and thus, the fluorescent signal can be taken as a proxy for proteolytic activity. Signal quantification showed a significant reduction in the fluorescence produced in KO microglia as compared to controls, indicative of a defective degradative capacity (Fig. 3a, b).

Phagocytic activity plays an important role in promoting the engulfment and the subsequent efficient degradation of synaptic elements[37], particularly in the early postnatal brain, when microglia critically contribute to sculpting neural circuits by elimination of redundant synaptic connections[38–42].

To understand whether the observed impaired degradation was also relevant to synaptic structures, we exposed microglial cells to fluorescently labeled synaptosomes isolated from CamKIIa^{CRE};Rosa26-fl-STOP-fl-TdTomato mice, and quantified their engulfment (at T0: after 1 hour incubation) and degradation (at T6: 6 hours following washout at T0). Confocal acquisition and imaging analyses revealed that MCT4 KO cells display a reduced engulfment of synaptosomes as well as a significant impairment in cargo degradation (Fig. 3c, d), consistent with their defective lysosomal acidification. Thus, we asked whether the functional response to lactate might be also altered in vivo, thereby affecting the microglia-mediated synapse remodeling in MCT4 cKO mice.

Lactate is particularly abundant in the brain in the perinatal period, consistent with key roles for this metabolite at early developmental windows[19,43]. Therefore, we set out to study the specific contribution of microglia to the lactate-driven effects on brain development, by means of the MCT4 cKO mouse line. Our investigation was focused on the hippocampus, which is a structure highly implicated in cognitive functions, and that undergo profound synapse refinement by microglia in the early postnatal weeks[38,40,42]. We measured lactate in the wildtype mouse brain at postnatal day (P)15, comparing its levels in the hippocampus to more rostral or caudal neocortical regions. Mass spectrometry analysis revealed that lactate is significantly more abundant in the hippocampus than in other neocortical areas, supporting a role for this metabolite in this structure, at least at this specific time point (Fig. 3e).

To address the role of microglial lactate transport in hippocampal development, we depleted MCT4 selectively in microglia by injecting tamoxifen at P6 and P8 in Cx3cr1^{CREert2/+};Slc16a3^{flox/flox} and Cx3cr1^{CREert2/+};Slc16a3^{+/+} male and female littermates, and brains were collected at P15 for downstream analyses (schematic representation, Fig. 3f). We confirmed the efficient KO of MCT4 in acutely isolated microglia, both at the transcript and at the protein level (Fig. 3g–i). Next, we asked whether the loss of MCT4 would affect microglial morphology. Three-dimensional reconstruction of hippocampal cells from control and cKO littermates did not show major structural alterations, as microglia from male and female subjects of both genotypes displayed similar Iba1+ volumes and surface area (Fig. 3j–l). While the number of CD68+ structures was also not affected, we observed a significant increase in their average volume (Fig. 3m, n). CD68 is a marker for phagolysosomes in microglia and macrophages, however, its presence cannot be directly taken as a proxy for phagocytic activity. Indeed, an increase in the volume of CD68+ structures could be an indication of augmented engulfment, or, on the contrary, it could point towards an excessive phagolysosomal accumulation, due to defective degradation. In light of our previous observations (Fig. 3a–d), we predicted that the increased size of phagolysosomes in MCT4 KO microglia may reflect the dysfunctional clearance of ingested cargoes, possibly including synaptic material.

## Defective synaptic pruning in MCT4 cKO mice

The early postnatal weeks represent a critical developmental window in the brain, in which synaptic pruning by microglia is particularly relevant to promote proper refinement of synaptic connections[44]. Thus, we hypothesized that MCT4 cKO mice, if faulty in microglia-mediated clearance, would show an accumulation of synaptic structures, particularly at this time. Thus, we performed Western blots on hippocampal homogenates from P15 mice, to screen for different pre- and post-synaptic markers, and found that a number of those (synapsin, VGAT, GluA2) were significantly more abundant in cKO mice than in control littermates (Fig. 4a, b). A significant increase in synapsin was further confirmed by immunostaining on hippocampal slices (Fig. 4c, d). Quantification of co-localizing puncta between the pre- and post-synaptic markers synapsin and homer1, respectively, revealed a significant increase in the number of synapses (Fig. 4c, e), supporting structural changes induced by the microglial loss of MCT4. Golgi-cox analysis indicated no changes in dendritic spine density between cKO

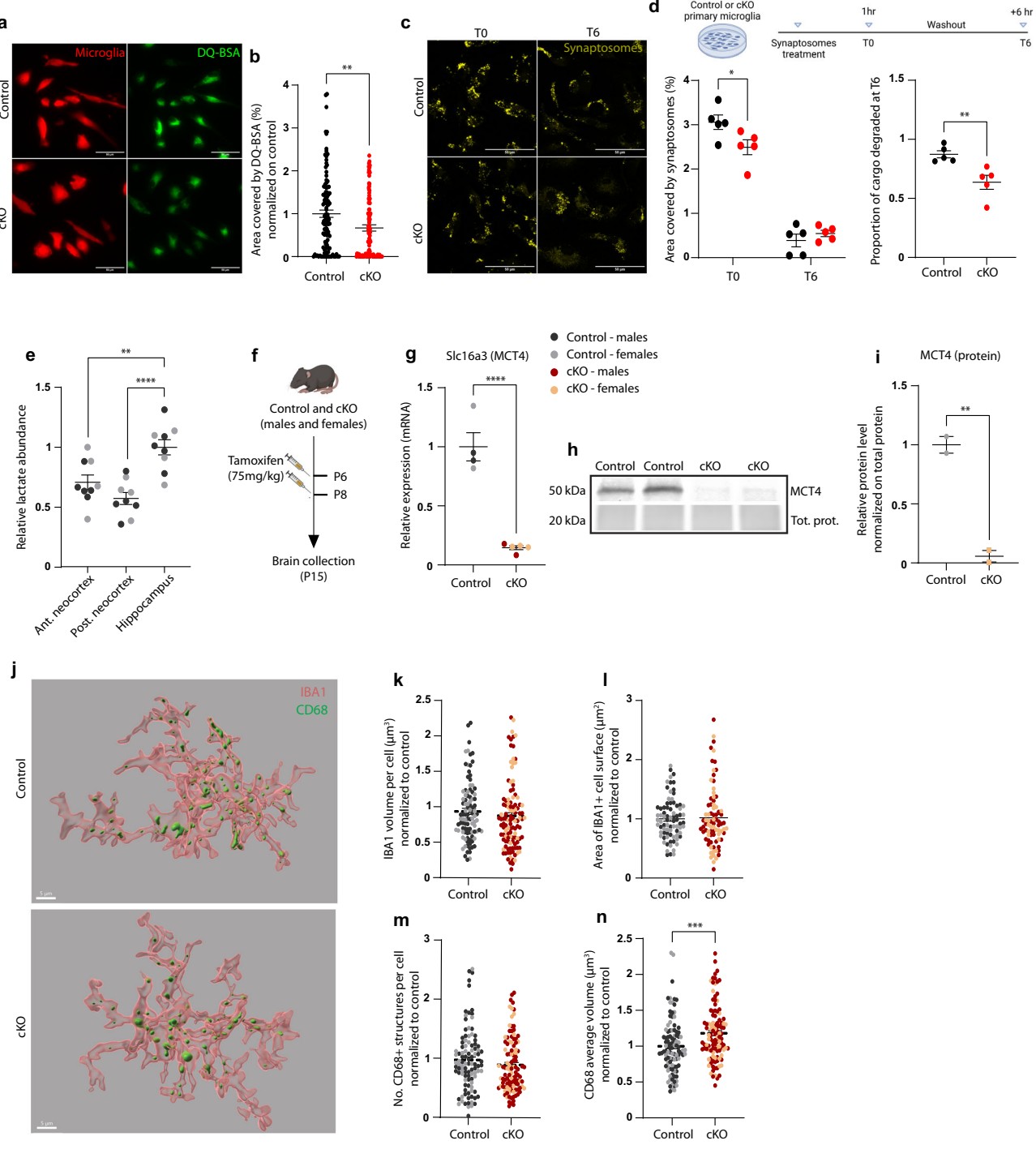

and control littermates (Fig. 4f, g), in line with no alterations in the levels of post-synaptic scaffold protein homer1 and PSD95 (Fig. 4a, b, Supplementary Fig. 4).

Overall, these data indicate that the selective loss of MCT4 in microglia not only causes intrinsic microglial dysfunction but also is sufficient to induce consequences at the synaptic level.

To further investigate whether the increase in synaptic markers was directly associated with defective microglial pruning in vivo, we assessed the engulfment of synapsin by microglia in the CA1 hippo-campal region of both control and cKO mice. The number of engulfed synaptic puncta was significantly higher in cKO mice (Fig. 4h, i). Fur-thermore, quantification of the total volume of synapsin within CD68+ phagolysosomal structures revealed a significant increase in cKO cells,

consistent with a defective degradation and consequent accumulation of the ingested material, as observed in vitro (Fig. 4h, j).

Overall, these data highlight a critical role for microglial MCT4 in mediating synapse elimination in the early postnatal hippocampus.

## Loss of microglial MCT4 produces synaptic adaptations and metabolic changes in the hippocampus

Next, we investigated whether the defective synapse remodeling in MCT4 cKO mice had functional repercussions on neurotransmission onto hippocampal neurons. We prepared acute brain slices from juvenile mice (P15 and P30) and performed patch-clamp in CA1 pyr-amidal cells. The amplitude of the recorded spontaneous excitatory post-synaptic currents (sEPSC) was significantly higher in cKO mice as

**Fig. 3 | MCT4 cKO microglia display reduced phagocytosis and lysosomal dysfunction in vitro and alteration of phagolysosomal structures in vivo.**
**a** Representative z-stack projections of microglia (TdTomato+) treated with DQ-BSA, and **b** relative quantification of area covered by cleaved DQ-BSA in control and cKO cells. Data point represent individual cells, control: $n = 111$ cells; cKO: $n = 103$ cells from $N = 2$ independent experiments. Two-tailed unpaired $t$ test, **$p = 0.0036$; Scale bar: 50 μm. **c** Representative confocal z-stack acquisitions of primary microglia exposed to fluorescently labeled synaptosomes (scale bar: 50 μm), and **d** relative quantification. Experiment timeline created with BioRender.com. *Left*: Microglial area covered by fluorescent signal per cell after 1 h exposure to synaptosomes (T0) and 6 h after medium washout (T6). Two-way ANOVA followed by Sidak's post hoc multiple comparison test, *$p = 0.0248$. Right: proportion of cargo degraded in 6 h in control and cKO cells. Two-tailed unpaired $t$ test, **$p = 0.0074$. Data points represent the average of at least 30 cells per condition, from $N = 5$ independent experiments. **e** Relative lactate abundance measured by LC-MS in different brain areas of P15 control mice. $N = 5$ males (black dots) and $N = 4$ females

(gray dots). One-way ANOVA followed by Dunnett's post hoc multiple comparison test; hippocampus vs. anterior neocortex: **$p = 0.0033$; hippocampus *vs.* posterior neocortex: ****$p < 0.0001$. **f** Schematic of experimental design for tamoxifen treatment and brain collection at P15. Created with BioRender.com. **g** Expression of Slc16a3 mRNA and **h, i** MCT4 protein in microglia acutely isolated from control and cKO mice at P15; for mRNA: Control: $N = 4$ ($N = 2$ males and $N = 2$ females) vs. cKO: $N = 5$ ($N = 2$ males and $N = 3$ females). For protein: $N = 2$ control females and $N = 2$ cKO females. Two-tailed unpaired $t$ test, ****$p < 0.0001$, **$p = 0.0082$.
**j** Representative 3D reconstruction of microglial cells from the CA1 *stratum radiatum* of control and cKO mice at P15, scale bar: 5 μm. Quantification of IBA1+ **k** cell volume, **l** surface area, **m** number of CD68+ structures within each microglia cell, and **n** average volume of CD68+ structures per cell, normalized to controls. **k–n** Data points represent single cells from $N = 4$ males and $N = 4$ females per genotype (P15); control: $n = 79–103$ cells; cKO: $n = 83–113$ cells. Two-tailed unpaired $t$ test, ***$p = 0.0007$. **b, d, e, g, i, k–n** Data are represented as mean ± SEM. Source data are provided as a Source Data file.

---

compared to controls, with a similar trend for sEPSC frequency (Fig. 5a–c). These results are indicative of a larger synaptic excitation onto CA1 pyramidal neurons, a phenomenon in accordance with the alterations observed at the level of synaptic proteins including synapsin and GluA2. Furthermore, the recording of AMPA- and GABA-mediated currents following electrical stimulation of the Schaffer collaterals in hippocampal slices from P15 mice revealed an increased AMPA/GABA ratio in pyramidal neurons from cKO mice compared to controls (Fig. 5d, e). Notably, microglial dysfunction and higher synaptic excitation onto CA1 pyramidal cells have been previously implicated in providing vulnerability to epileptic seizures[38,45,46]. We therefore predicted that MCT4 cKO mice would be more susceptible to epileptogenic drugs. To this aim, we administered a single sub-threshold dose of kainic acid (KA), which is a potent neuroexcitatory compound used to study mechanisms of hippocampal-dependent seizures by mimicking glutamatergic neurotransmission[47]. As expected, intraperitoneal administration of 2 mg/kg KA did not cause any seizure-related response in control mice (Fig. 5d, e). However, 100% of the cKO males and females developed clonic seizures as early as one hour upon injection (Fig. 5d, e).

These data confirm that the loss of microglial MCT4 leads to synapse structural and functional alterations in the hippocampus of juvenile mice.

To further characterize global changes in the hippocampus at the metabolic level, we performed LC-MS untargeted metabolomics on hippocampal homogenates extracted from control and cKO littermates at P15. Increase in 2,3-biphospho-D-glycerate, phosphoenolpyruvate, and in pyruvate in cKO hippocampi supported possible alterations in the glycolytic pathway (Supplementary Fig. 5). Analysis of the tricarboxylic acid (TCA) cycle revealed a significant increase in the intermediate metabolite succinate (Fig. 5f), consistent with a role for its accumulation in promoting epileptic seizures[48]. Finally, cKO hippocampi presented increased levels of diverse acylcarnitine intermediates, such as hydroxybutyrylcarnitine, hexanoyl-L-carnitine, L-octanoylcarnitine and hexadecenoyl-carnitine (Fig. 5g–k). Acylcarnitines (ACs) are formed from carnitine and acyl-CoAs by carnitine acyltransferases in mitochondria or peroxisomes[49]. Since the metabolism of fatty acids, glucose, and amino acids can yield ACs, the high levels found in cKO mice might be indicative of a metabolic adaptation, in the absence of microglial MCT4.

### Adult cKO mice display anxiety-like behavior

Defective microglia-mediated circuits remodeling during brain development often results in long-term behavioral consequences[50]. Thus, to investigate whether early postnatal depletion of MCT4 in microglia could lead to behavioral alterations in adulthood, we performed several tests aimed at characterizing different domains of mouse behavior.

We confirmed that MCT4 depletion in microglia was long lasting, with significant reduction in cKO persisting in adulthood (reduction of transcript level, Supplementary Fig. 6a).

Mice were first tested in the open field arena to assess basic locomotor activity as well as anxiety-like behavior. While locomotion and immobility time were comparable between sexes and genotypes (Supplementary Fig. 6b, c), adult cKO males spent significantly less time exploring the center of the arena, and traveled reduced distance in the center, as compared to control littermates, indicative of increased anxiety (Fig. 6a–c).

This was confirmed in the elevated plus maze (EPM) test, in which cKO males spent significantly more time in the closed arms (Fig. 6d), where they also exhibited increased immobility time (Fig. 6e). While the anxiety phenotype was sex-dependent and only present in males, females displayed a different abnormal behavior, with significantly reduced exploration and average speed in the EPM (Fig. 6f, Supplementary Fig. 6d, e). Furthermore, we analyzed self-grooming behavior, both in response to a novel environment and in response to physical stimulation (i.e., water spray on the snout). While cKO males displayed similar levels in both types of grooming as compared to controls, as well as the expected increase upon water spray, cKO females showed a significant reduction in the spray-induced grooming, consistent with freezing response (Fig. 6g, h). Finally, we also tested motor performance as well as spatial memory and working memory using rotarod (Fig. 6i), Barnes maze (Fig. 6j, k), and Y maze (Fig. 6l) tests, respectively. We found no differences in any of these tests, suggesting that spatial memory and motor coordination are not affected by the loss of microglial MCT4.

Quantification of synaptic markers in the adult hippocampus revealed a remarkable loss of both pre- and post-synaptic markers, such as VGLUT1, PSD95, Gephyrin, and GluA2 in male and female cKO mice (Fig. 7a, b). Reduction in PSD95 was further confirmed by immunostaining in hippocampal slices, indicating prominent synapse loss in adulthood.

Overall, these data support long-term alterations in the anxiety-like behavior of cKO mice, accompanied by structural synaptic alterations, indicating that microglial MCT4 depletion is sufficient to affect complex brain functioning and synapse integrity in adulthood.

## Discussion

In this study, we investigated the role of the monocarboxylate transporter 4 (MCT4) in the modulation of microglial function and showed that microglia lacking MCT4 display defective lysosomal acidification and impaired synaptic pruning, with long-lasting consequences on synaptic function and mouse behavior.

The high expression of different metabolic signatures in microglia represents the basis for their metabolic flexibility, supporting a rapid metabolic adaptation according to the availability of energetic

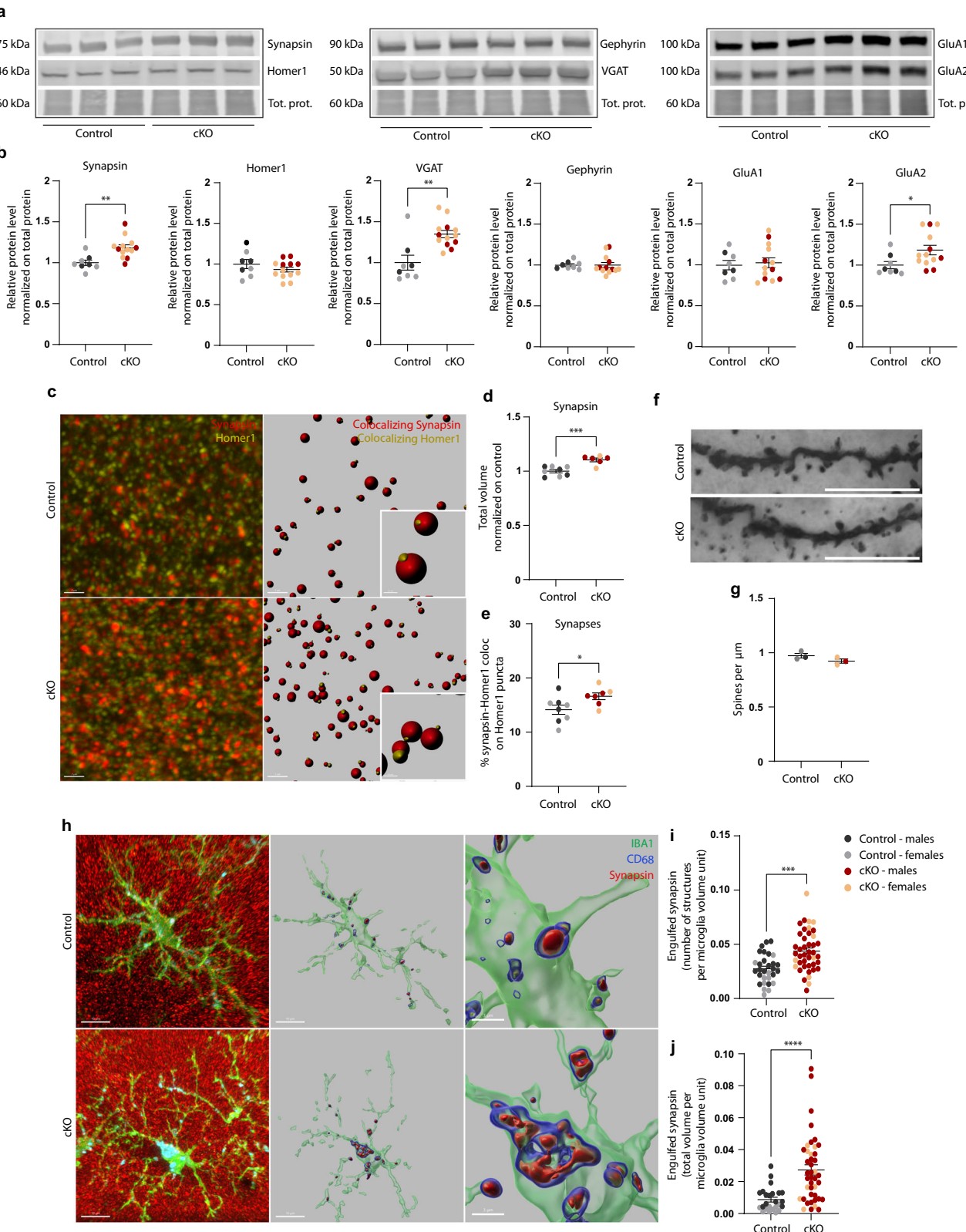

substrates. While glucose has been considered for long time the major bioenergetic fuel for microglia, little is known about the utilization of alternative metabolites. The high microglial expression of the *LDHB* gene, which is preferentially expressed by cells converting lactate into pyruvate, led us to hypothesize that microglia might efficiently metabolize lactate. A growing body of literature now established lactate as a key metabolite, which not only can be a primary source of carbon for the TCA cycle[51,52], but it also contributes to determining the cellular NADH/NAD+ ratio sustaining redox buffering[53,54].

Oxidation of lactate, requiring NAD+, generates NADH and protons, which promote V-ATPase-dependent lysosomal acidification[32]. Given the prominent role of lysosomal degradation in microglia, we set out to study whether lactate plays any role in modulating microglial function.

**Fig. 4 | Deletion of MCT4 in microglia leads to altered synaptic pruning in the hippocampus. a** Representative western blots of synaptic markers in hippocampal homogenate from P15 control and cKO littermates and **b** relative quantification, normalized to the control group. Each data point corresponds to one animal. Control: $N = 3$ males; $N = 5$ females; cKO: $N = 5$ males; $N = 8$ females. Two-tailed unpaired $t$ test, $**p$(synapsin) = 0.0024; $**p$(VGAT) = 0.0011; $*p$(GluA2) = 0.0372. **c** Representative confocal z-stack projections and relative 3D reconstruction of synapsin- and Homer1-positive puncta in the CA1 *stratum radiatum* from control and cKO P15 mice. Scale bar: 2 μm and 0.3 μm in the inset. **d** Quantification of the total volume of synapsin-positive puncta measured in the CA1 *stratum radiatum* and dentate gyrus of control and cKO mice at P15, normalized to controls. Each data point corresponds to one animal. Control: $N = 4$ males; $N = 4$ females; cKO: $N = 4$ males; $N = 2$ females. Two-tailed unpaired $t$ test, $***p = 0.0007$. **e** Excitatory synapses quantification in the CA1 *stratum radiatum* of control and cKO mice at P15. Individual synapses are defined as the colocalization of the pre- and post-synaptic

markers synapsin and Homer1. Each data point corresponds to one animal. Control: $N = 4$ males; $N = 4$ females; cKO: $N = 4$ males; $N = 3$ females. Two-tailed unpaired $t$ test, $*p = 0.0412$. **f** Representative dendrites of CA1 pyramidal neurons in control and cKO mice at P15, visualized by Golgi-cox staining. Scale bar: 10 μm. **g** Quantification of spine density reported as absolute numbers per μm. Each data point represents one animal, as the average of $n = 3$–18 dendrites segments. Group size $N = 3$ ($N = 1$ male and $N = 2$ females per genotype). **h** Representative confocal z-stack projections and relative 3D surface reconstruction of IBA1-positive microglia cells, engulfing synapsin puncta within CD68-positive phagolysosomal structures, from control and cKO mice (P15, CA1 *stratum radiatum*). Left panel: scale bar 10 μm; Zoom-in: scale bar 3 μm. **i, j** Number and total volume of engulfed synapsin puncta per microglia volume unit. Two-tailed unpaired $t$ test, $***p = 0.0001$; $****p < 0.0001$. Data points represent individual cells from $N = 3$ males and $N = 4$ females per genotype; $n = 30$ control cells, $n = 42$ cKO cells. **b, d, e, g, i, j** Data are represented as mean ± SEM. Source data are provided as a Source Data file.

By using fluorescent analogs of lactate and mass spectrometry, we show that extracellular lactate can be imported into microglia, and it functionally controls lysosomal activity, by decreasing late endosomes/lysosomes pH. The lactate-induced effect on lysosomes is completely abolished both in the presence of the V-ATPase inhibitor bafilomycin and in MCT4 KO cells, indicating that this mechanism relies on the lysosomal proton pump, and on the lactate transporter MCT4. Recent data show that MCT4 mediates lactate uptake even at low concentrations, with a $K_m$ of 1.7 mM, establishing MCT4 as a high-affinity lactate transporter[55]. Furthermore, our data indicates that MCT4 is the only lactate transporter, among the ones highly expressed in the brain, which is significantly upregulated in microglia in presence of extracellular lactate, suggesting its functional implication for the import of this metabolite. Dose titration experiments also revealed that lactate-induced lysosomal modulation is observed starting at a concentration of 3 mM. In the normal brain, the concentration of extracellular lactate has been estimated to range between 1 mM and 3 mM, thus supporting a physiological regulation of this process[56]. Furthermore, microdomain of elevated lactate concentration are plausible to occur at localized synapses, with possible space-dependent modulation of microglial function.

By crossing the Cre mouse line Cx3cr1[CREert2] with Slc16a3 floxed mice, we generated and characterized a cKO line in which only microglia and macrophages lack MCT4, upon tamoxifen administration. This study aims at disrupting in vivo the transport of lactate in microglia. It is important to emphasize that MCT4 is responsible for the bidirectional transport of lactate across the plasma membrane. While we focus the entire study on the effects of lactate import in microglia in a context of high lactate availability (i.e., early postnatal hippocampus), we are aware that disrupting MCT4 could also lead to lactate intracellular accumulation in specific circumstances. For instance, if cells become hyper glycolytic, as in the case of infection or in neurodegeneration, it is plausible to speculate that high lactate production will result in intracellular lactate accumulation, when MCT4 is depleted. In support of this, previous in vitro studies have shown that MCT4 knockdown leads to augmented levels of intracellular lactate and decreased glycolysis in LPS-treated macrophages[57]. Therefore, the role of MCT4 and lactate transport in microglia in a pathological context warrants further investigations.

In this study, we focused on microglial MCT4 in healthy brain development and found that impaired degradation capacity of MCT4 KO microglia results in defective synaptic pruning, associated with structural and functional alterations of hippocampal neurons. While the pre-synaptic marker synapsin was found to accumulate in the hippocampus and within the microglia of cKO mice at P15, no changes were observed for post-synaptic scaffold proteins, such as PSD95 and homer1. Dendritic spine density analysis further confirmed no changes in the number of post-synaptic structures. However, the levels of AMPAR subunits were increased in cKO mice, suggesting an effect on

the composition of post-synaptic terminals. Overall, our findings substantiate previous literature showing that microglia-mediated synaptic refinement is a complex process, which can differentially affect pre- and post-synaptic structures, as well as subset of synapses over others, largely depending on the timing and triggers[58].

Microglia can control the number of functional synapses and modulate their activity, and several studies have causally linked dysfunctional microglia to altered synaptic function[38,42,59,60]. We found that pyramidal neurons in the CA1 area of the hippocampus of juvenile cKO mice received an enhanced excitatory drive and display increased AMPA/GABA ratio. Altogether, these finding corroborate a scenario in which cKO mice present strengthening of post-synaptic excitatory neurotransmission, although adaptations at the level of GABA receptors cannot be ruled out. These alterations could be explained by defective microglia-mediated synapse refinement. Treatment of hippocampal slices with the macrophage inhibitor factor MIF was shown to induce an increase in the amplitude of sEPSCs, without affecting their frequency, similarly to our findings[60]. Interestingly, MIF significantly diminish the phagocytotic activity of microglia[61], in line with our MCT4 KO phenotype. Additionally, Peng and colleagues[62] reported that knocking out P2ry12, a microglia-specific receptor involved in microglia-synapse interaction and modulation of synaptic activity[63,64], leads to increased amplitude of sEPSCs in CA1 pyramidal neurons, coupled with enhanced innate fear behaviors, consistent with what we have observed in MCT4 cKO mice. We could also hypothesize that the lysosomal defects observed in MCT4 KO microglia, in addition to phagocytosis, may affect other fundamental cellular processes, such as motility and production or release of soluble factors. Therefore, in combination with structural defective pruning, this may contribute to a deficient regulation of synaptic activity and function. However, this remains to be investigated in future studies. Another possibility is that inhibition of lactate intake by MCT4 KO microglia causes an elevated extracellular concentration of lactate, which could become more available to neighboring cells, thus contributing to the modulation of the activity of pyramidal neurons.

Increased E/I ratio and dysfunctional microglia have been previously linked to pathophysiology of epilepsy[45,65–67]. Here we found that loss of microglial MCT4 promotes vulnerability to epileptic seizures in a model of sub-threshold doses of KA. Importantly, succinate accumulation has been shown to be sufficient to induce status epilepticus, similar to those observed in KA-induced subjects[48]. Thus, the increased succinate accumulation we found in the hippocampus of cKO mice might contribute to the increased susceptibility to KA. Of note, it was shown in the heart that MCT1 is the succinate transporter allowing it to be released and act on neighboring cells via a specific G-coupled receptor[68]. Whether MCT4 in microglia could play such a role is still unknown. Furthermore, we reported a significant increase in a number of acylcarnitine intermediates, in line with metabolic adaptation following microglial MCT4 depletion. Yet, our metabolomics

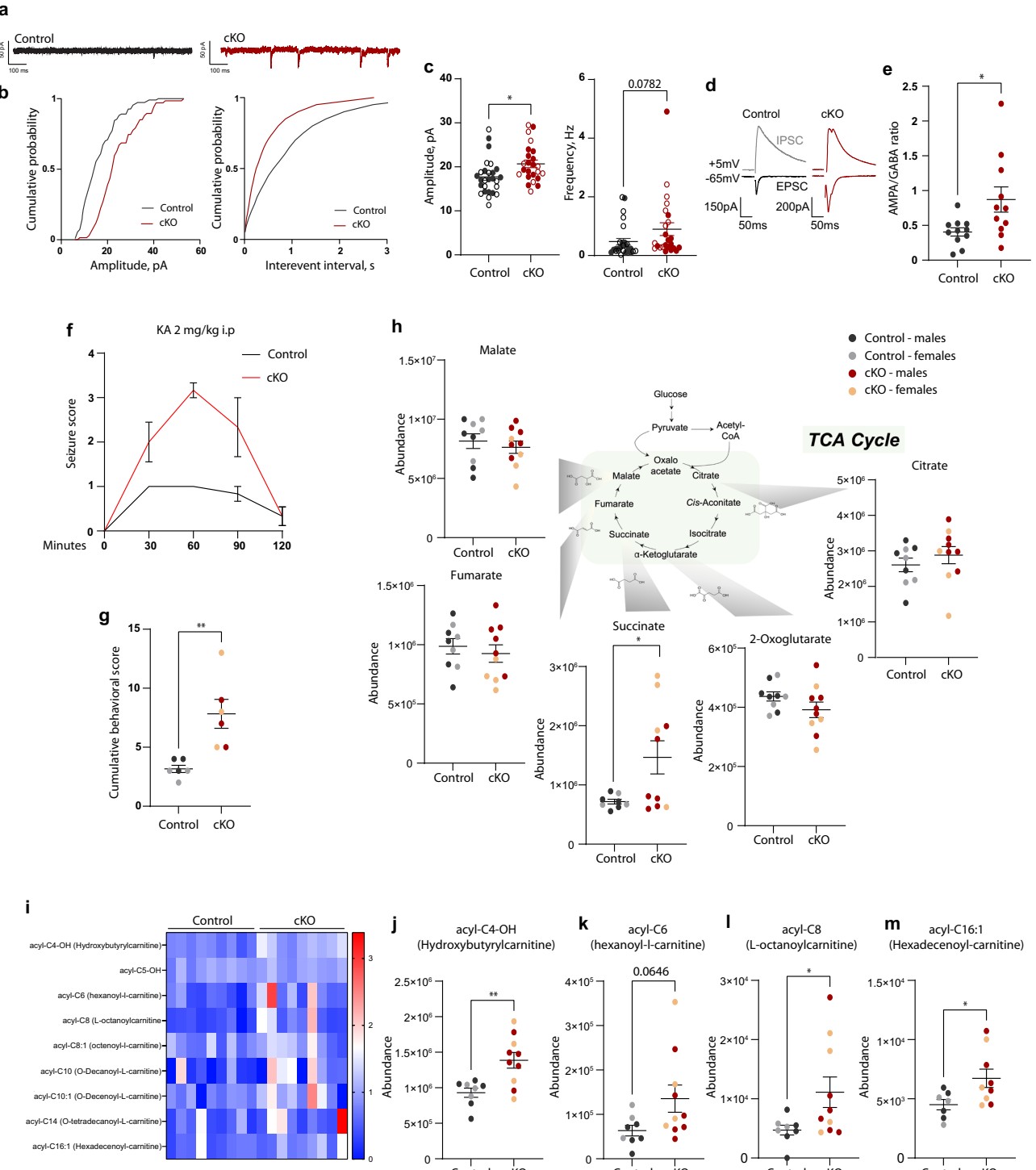

**Fig. 5 | Loss of microglial MCT4 induces functional synapse alterations and metabolic changes in the hippocampus. a–c** Whole-cell recordings of CA1 pyramidal cells in brain slices from control and cKO male littermates at P15 and P30. **a** Representative traces from control and cKO mice. **b** Representative cumulative distributions of spontaneous excitatory post-synaptic currents (sEPSCs) amplitude and frequency. **c** Average values of sEPSCs amplitudes and frequencies (controls: $n = 27$ cells from $N = 2$ P15 and $N = 3$ P30 male mice; cKO: $n = 24$ cells from $N = 2$ P15 and $N = 2$ P30 male mice; two-tailed unpaired $t$ test; *$p = 0.0126$). Data points correspond to individual cells. Filled data points indicate cells recorded at P15; empty data points are for cells analyzed at P30. **d, e** AMPA/GABA ratio of CA1 pyramidal neurons measured in P15 control and cKO male littermates. **d** Representative traces of excitatory (AMPA) and inhibitory (GABA) post-synaptic currents (EPSCs and IPSCs) recorded from control and cKO mice and **e** relative quantification. Data points correspond to individual cells. Control: $n = 11$ cells, cKO: $n = 11$ cells from

$N = 2$ mice per genotype; two-tailed unpaired $t$ test; *$p = 0.0241$. **f, g** Susceptibility to kainic acid-induced seizures at P15; $N = 3$ males and $N = 3$ females per genotype. **f** Seizure score according to the modified Racine scale, plotted as maximum score reported for each mouse in a 30 min period. A score of 3 corresponds to the development of clonic seizures. **g** Cumulative behavioral score per mouse for the total observation period (120 minutes). Two-tailed unpaired $t$ test, **$p = 0.0041$. **h–m** Abundance of metabolites measured via LC-MS in the hippocampus of P15 mice. Control: $N = 5$ males; $N = 4$ females. cKO: $N = 6$ males; $N = 4$ females. **h** TCA cycle metabolites. Two-tailed unpaired $t$ test, *$p$(succinate) = 0.0312; **i** Heatmap representing the relative abundance of carnitine-related metabolites in control and cKO mice at P15. **j–m** **$p$(acyl-C4-OH) = 0.0043; *$p$(acyl-C8) = 0.0472; *$p$(acyl-C16:1) = 0.0358; two-tailed unpaired $t$ test. **c, e, f–m** Data are represented as mean ± SEM. Source data are provided as a Source Data file.

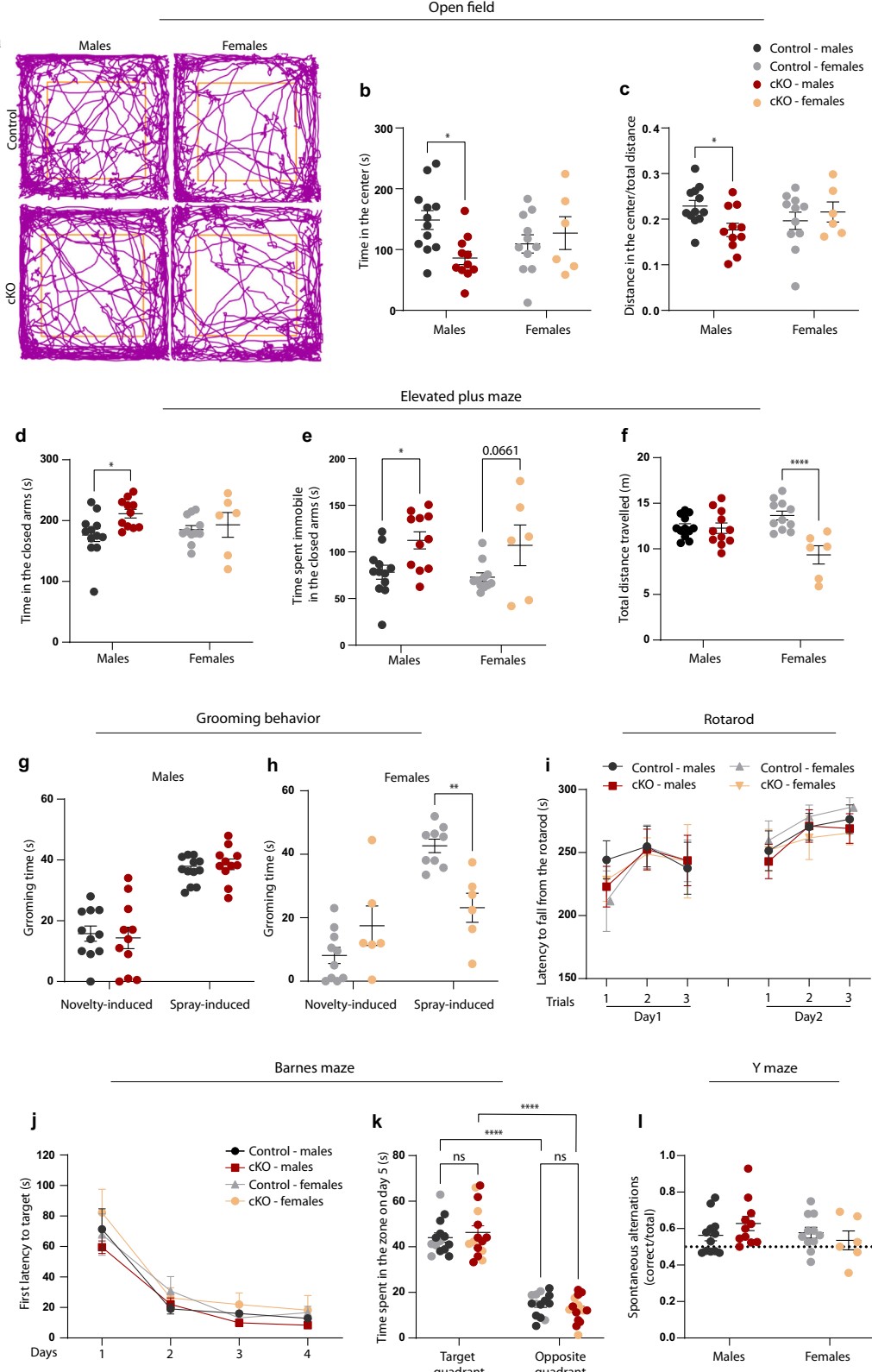

study has been performed on whole hippocampal tissue, therefore making it impossible at this stage to distinguish microglia-specific changes from metabolic adaptations in neighboring cells.

Different signaling mechanisms have been described for microglia to refine synaptic connections, such as the complement system, the Cx3cr1/Cx3cl1 signaling pathway, the purinergic receptor P2Y12, and the TREM2 receptor[38–42,44,59,69]. All these mechanisms are engaged in a context-dependent manner, as they are active in certain brain regions but not in others, in specific developmental windows. Whether MCT4-mediated microglial contribution to synaptic refinement is relevant only in the hippocampus, or also in other brain structures, it remains to be addressed.

Defective synaptic refinement by microglia in the early postnatal period has been shown to lead to long-lasting consequences on brain

**Fig. 6 | Adult cKO mice display an anxiety-like behavior.** Behavioral characterization of control and cKO adult mice (7–8 months). Each data point represents one animal. Control: $N = 12$ males; $N = 11$ females; cKO: $N = 11$ males; $N = 6$ females. Statistics are calculated with two-way ANOVA followed by Sidak's post hoc multiple comparison test. **a–c** Open field exploration (20 minutes), with **a** representative tracking traces; **b** quantification of time spent in the center of the arena; *$p$(males) = 0.0105; **c** fraction of distance traveled in the center; *$p$(males) = 0.0418. **d–f** Elevated plus maze test (5 minutes), with quantification of **d** time spent in the closed arms, *$p$(males) = 0.0279; **e** time spent immobile in the closed arms, *$p$(males) = 0.0216; **f** total distance traveled in the maze; ****$p$(females) <0.0001.

**g, h** Quantification of grooming time induced by a novel cage (novelty-induced) and upon physical stimulation (water spray on the snout, spray-induced) in **g** males and in **h** females; **$p$(females) = 0.0019. **i** Latency to fall from the accelerating rotarod. Each mouse underwent three trials per day, on two consecutive days. **j, k** Barnes maze test, consisting of 4 days training, four trials per day, followed by a single trial with no escape box, on day 5. **j** First latency to reach the escape hole. **k** Time spent in the quadrant where the escape hole was located (target quadrant) and at the opposite side of the maze on the test day. ****$p$(target-opposite) <0.0001. **l** Proportion of correct spontaneous alternation in the Y maze test. **b–l** Data are represented as mean ± SEM. Source data are provided as a Source Data file.

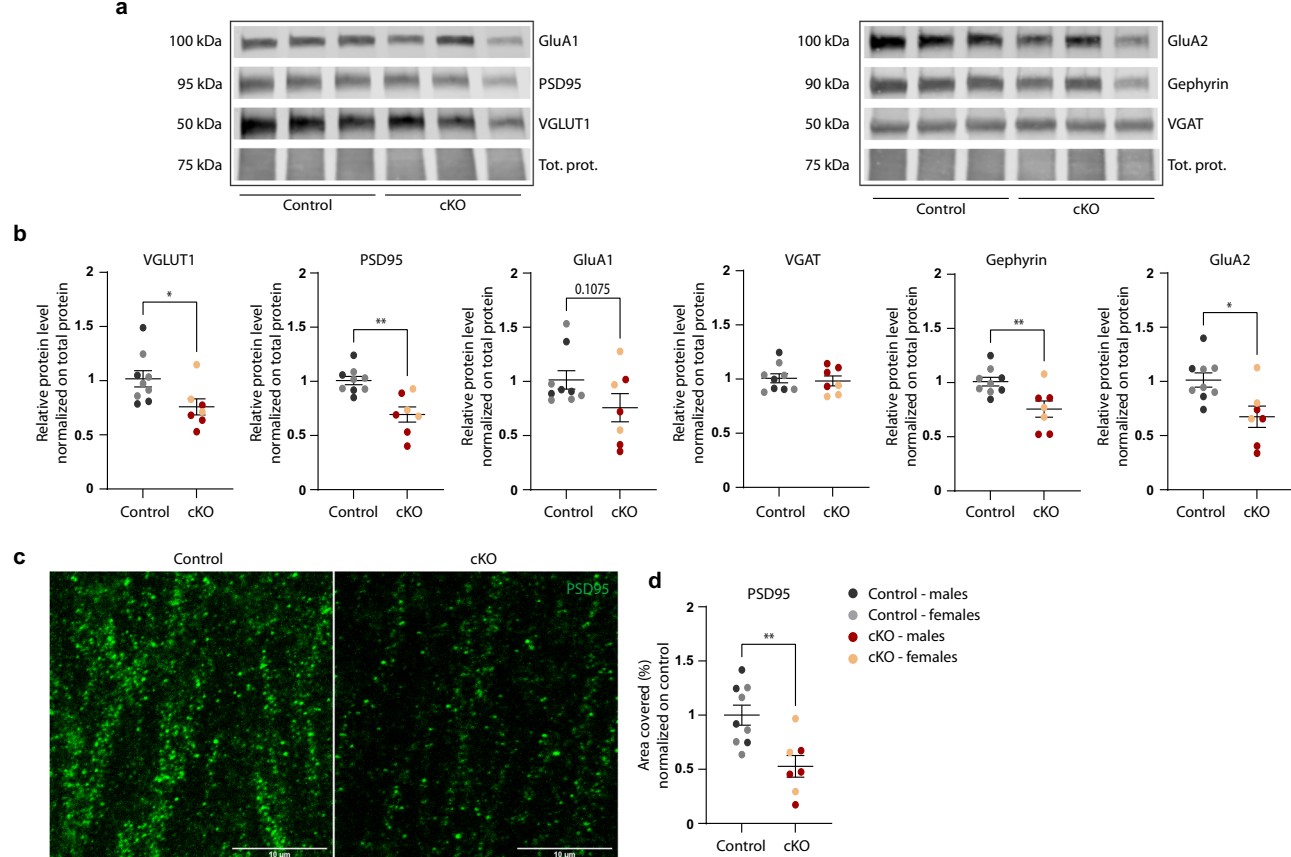

**Fig. 7 | Adult cKO mice display a significant decrease in the content of synaptic markers in the hippocampus. a** Representative western blots of synaptic markers in hippocampal homogenate from 7–8 months old control and cKO littermates and **b** relative quantification, normalized to the control group. Each data point corresponds to one animal. Control: $N = 4$ males; $N = 5$ females; cKO: $N = 4$ males; $N = 3$ females. Two-tailed unpaired $t$ test, *$p$(VGLUT1) = 0.0307; **$p$(PSD95) = 0.0010; **$p$(Gephyrin) = 0.0072; *$p$(GluA2) = 0.0101. **c** Representative confocal $z$-stack

projections of PSD95 puncta in the CA1 *stratum radiatum* from adult control and cKO mice (7–8 months). Scale bar: 10 μm. **d** Quantification of the area covered by PSD95-positive puncta, normalized to controls. Each data point corresponds to one animal. Control: $N = 4$ males; $N = 5$ females; cKO: $N = 4$ males; $N = 3$ females. Two-tailed unpaired $t$ test, **$p$ = 0.0038. **b, d** Source data are provided as a Source Data file.

connectivity and behavior[40,42]. Additionally, microglia have been causally linked to the development of anxiety-like behavior[62,70]. In our study, we reported long-lasting effects of MCT4 depletion in microglia, as adult cKO subjects display increased anxiety. Additionally, we observed that several behavioral phenotypes in cKO mice were sex-dependent. We found that male cKO mice display a classical anxiety-like behavior, as they preferentially explore more protected areas of the open field and elevated plus maze, i.e., the corridors and the closed arms, respectively. Even though this phenotype was not present in cKO females, they display reduced exploration and increased immobility in the elevated plus maze, two behaviors that are also associated with augmented anxiety in mice. Males and females have been reported to display significant differences in locomotion and differential

compartment occupancy in the elevated plus maze test[71,72]. Furthermore, collective evidence indicates that sex differences are prominent in anxiety disorders[73], which may explain the sexually dimorphic behavior we observed in MCT4 cKO mice.

However, it is important to note that at the cellular level, we did not observe major differences between male and female cKO mice, indicating that the manifestation of sexually dimorphic behavior implies more complex mechanisms, which might include differences in the underlying neural circuitries and in the hormonal regulation[74]. Despite synaptic alterations in the hippocampus, cKO mice did not display deficits in spatial and working memory, suggesting that the circuits affected may preferentially involve ventral hippocampus-amygdala connectivity, more relevant for anxiety and stress[75]. While

synaptic structures were exuberant in the hippocampus of cKO mice at early developmental stages, they drastically decreased in adulthood, including markers of pre-synaptic terminals and the scaffold post-synaptic density protein PSD95. It may be speculated that weak and exuberant synapses are more prone to be lost across the lifespan, or that defective synaptic growth leads to reduced density over time, as previously observed in TREM2 KO mice[42]. It cannot be excluded that additional microglial MCT4-dependent processes are implicated in homeostatic maintenance of synapse integrity in adulthood. Dissecting the exact molecular mechanisms underlying microglia-synapse interaction in MCT4 cKO mice will be the goal of future studies.

In summary, we show that microglial lactate import regulates lysosomal acidification through an MCT4-dependent mechanism. Selective microglial depletion of MCT4 leads to impaired phagocytosis and altered synaptic pruning. Functionally, this is associated with increased excitation in hippocampal neurons, enhanced vulnerability to kainic acid-induced seizures, as well as with the onset of an anxiety-like phenotype in adult cKO animals.

## Methods

### Animals
All animal experiments were authorized by the Service de la consommation et des Affaires vétérinaires (SCAV) of the Canton de Vaud in Switzerland, and by the Direction générale de l'agriculture, de la viticulture et des affaires vétérinaires (DGAV) - Affaires vétérinaires Protection des animaux, under the animal license VD3817. Mice were bred and maintained in the animal facility of the University of Lausanne. Cx3cr1CRE$^{ERT2}$;MCT4$^{flox}$ mice were obtained by crossing the B6.129P2(Cg)-Cx3cr1$^{tm2.1(cre/ERT2)Litt}$/WganJ mice (Cx3cr1CRE$^{ERT2}$; No: 021160, The Jackson Laboratory) with C57Bl/6.MCT4$^{tmlflox}$ mice (MCT4$^{flox}$; kindly provided by Prof. Luc Pellerin). For specific experiments, the Cx3cr1CRE$^{ERT2}$;MCT4$^{flox}$ line was further crossed with B6.Cg-Gt(ROSA)26Sor$^{tm14(CAG-tdTomato)Hze}$/J mice (TdTomato$^{flox}$; No: 007914, The Jackson Laboratory).

Mice were group-housed and kept in a 12 h day/night cycle, with food and water *ad libitum*, at 20–22 °C. All the experiments were performed during the day cycle. Both male and female mice were analyzed, unless differently specified. For in vivo gene KO induction, tamoxifen (cat no T5648, Sigma Aldrich) was prepared in 10% ethanol in corn oil at a concentration of 6 mg/ml. The whole litter was injected intraperitoneally (i.p.) at P6 and P8 (75 mg/kg). For biochemical and histological assessments, mice were terminally anesthetized with sodium pentobarbital diluted in saline (150 mg/kg, i.p.) and perfused with cold HBSS (cat no 14175129, Life Technologies, 3 ml/min). Upon brain collection, the left hemisphere was postfixed in cold PFA 4% for 24 h and stored at 4 °C in 0.02% of NaN$_3$ in PBS. The hippocampus was dissected from the right hemisphere, immediately frozen in dry ice, and stored at −80 °C.

### Microglia primary cultures
Primary microglia cells were obtained as previously described[76], with slight modifications. Briefly, mouse pups at P3–5 were sacrificed by decapitation and the brain was collected in HBSS. Olfactory bulb and cerebellum were removed, and the meninges were peeled off. The tissue was grossly minced, transferred into TripLE Express Enzyme Solution (cat no 12-604-021, Life Technologies), and incubated at 37 °C for 20 min followed by mechanical dissociation. DMEM high glucose (cat no 41966052, Gibco), supplemented with 10% FBS and 1% pen-strep (PS) was added to the preparation at a 3:1 ratio. Cells were pelleted at 400 × *g* for 4 min at RT. The pellet was resuspended in fresh culture medium and plated in a T75 cell culture flask. Cultures were maintained at 37 °C and 5% CO$_2$. After ca. 2 weeks, microglia were detached from the underlying astrocytes layer by smacking the flask. Cells were pelleted at 350 g for 5 min, resuspended in astrocytes-conditioned medium (ACM), and seeded on poly-D-lysine (PDL)-coated plates. All following treatments and medium changes were performed using ACM.

### SCOTfluor-based lactate import assay
The fluorescently labeled L-lactate (SCOTfluor)[30] was kindly provided by Prof. Marc Vendrell, University of Edinburgh (UK). Primary microglia cells were pre-treated for 1 h with either control (ACM) or with low glucose-containing medium (0.1 mM). The SCOTfluor lactate was diluted in control or low glucose-medium to a concentration of 1 mM and applied onto primary microglia for 1 h. Imaging was performed using a stage incubator (37 °C, 5% CO$_2$) on a LSM710 (Zeiss, ×40, Z-stack size 5.5 μm, 0.5 μm step size).

### Mass spectrometry (MS)-based lactate import assay
Primary microglia were treated for 6 h with either control medium (DMEM high glucose, 10% dialyzed FBS, 1% PS) or with 20 mM sodium lactate in control medium. At the end of treatment, cells were washed with PBS, and plates were stored at −80 °C. Cell lysates were quenched and extracted with methanol 80%. Homogenates were prepared using the Cryolys Precellys Homogenizer (Bertin Technologies), with ceramic beads. Homogenized extracts were centrifuged at 4 °C for 15 min at 4000 × *g* and supernatants were analyzed by HILIC-HRMS on a Vanquish Horizon (Thermo Fisher Scientific) ultra-high performance liquid chromatography (UHPLC) system coupled to Q Exactive™ Focus interfaced with a HESI source. Chromatographic separation was carried out using a SeQuant ZIC-pHILIC column (Merck). Full scan HRMS acquisition mode (m/z 50−750) was used at resolution at 70,000 FWHM, 1 microscan, 1e6 AGC and 100 ms as maximum inject time. Data were processed using Xcalibur (version 4.1, Thermo Fischer Scientific). Relative quantification of metabolites was based on EIC (Extracted Ion Chromatogram) areas for the monitored full scan peaks.

### In vitro treatments
4-hydroxytamoxifen (cat no H7904, Sigma Aldrich, 500 μM in 2.5% ethanol in PBS) was diluted in ACM to a concentration of 500 nM and was applied onto primary microglia cultures, with medium change after 24 h. Imaging-based assays or cell lysis were performed after 5 days. For lactate treatments, primary microglia were treated with either ACM control or ACM supplemented with sodium lactate (25 mM, unless indicated differently) for 24 h. Cells were then either lysed for biochemical studies or further utilized for imaging-based assays. For pyruvate treatment, media containing 5 mM, 10 mM and 25 mM sodium pyruvate were obtained starting from a 100 mM stock solution (cat no S8636, Sigma Aldrich). Dilutions were prepared fresh the day of the experiment and applied onto cells for 24 h. To inhibit v-ATPase, bafilomycin A1 (cat no HB1125, HelloBio) was reconstituted in DMSO at the concentration of 100 μM. The day of the experiment, the stock concentration was diluted 1:1000 (final: 100 nM) and cells were pre-treated for 3 h before undergoing further procedures. Where appropriate, bafilomycin was kept in the medium during the whole duration of the following cellular assays.

### Lactate quantification in astrocytes-conditioned medium (ACM)
Lactate concentration in the cell culture medium was measured using an enzymatic-spectrophotometric method. Briefly, cell culture medium was diluted 1:100 and compared to a standard of known lactate concentrations, ranging from 200 μM to 0 μM (blank). Reactions containing LDH enzyme, NAD$^+$ and cell culture medium were prepared in duplicate as described by Rosenberg and Rush[35].

### In vitro functional assays (LysoTracker, DQ-BSA and synaptosomes phagocytosis)
For labeling acidic organelles, primary microglia were incubated for 1 h with LysoTracker Red DND-99 (cat no L7528, Life Technologies, 200 nM). For assessing lysosomal proteolytic capacity, cells were

incubated for 1 h with DQ Green BSA (cat no D12050, Life Technologies, 100 μg/ml). Both compounds were diluted in ACM or treatment medium, where appropriate. Cells were washed with PBS and fixed (PFA 4%, 37 °C, 20 min). Nuclei were stained for 5 min at RT with DAPI (1 μg/ml in PBS) and coverslides were mounted in Mowiol 4–88 (cat no 81381, Sigma Aldrich). For synaptosomal phagocytic assay: synaptosomes were isolated from adult CamKIIa[cre/+];Rosa26-fl-STOP-fl-TdTomato[77,78] brains using Syn-PER™ Synaptic Protein Extraction Reagent (cat no 87793, Life Technologies) according to manufacturer's instructions. The synaptosomes-containing pellet was resuspended in 5% DMSO in Syn-PER™ to obtain a concentration of 6.5 μg/ml. Aliquots were stored at −80 °C. Primary microglia were incubated for 1 h with synaptosomes (65 μg/ml in ACM) and either fixed, stained and mounted as above, or extensively washed with ACM and fixed 6 h after. Acquisitions were taken at a Stellaris 5 confocal microscope (Leica, ×63 objective, Z-stack size 5.05 μm, step size 0.3 μm).

## Metabolomics of brain tissue

Mice (P15) were sacrificed by decapitation and the brain regions of interest were quickly dissected and snap frozen in liquid nitrogen. Specifically, frozen tissue samples were extracted at 20 mg/mL using ice-cold 5:3:2 methanol:acetonitrile:water. Samples were homogenized using a bead beater for 5 min, on ice. The homogenates were then vortexed 30 min and spun down for 10 min at 18,000 g at 4 °C. After sample randomization, 10 μL of extracts were injected into a Thermo Vanquish UHPLC system (San Jose, CA, USA) and resolved on a Kinetex C18 column (150 × 2.1 mm, 1.7 μm, Phenomenex, Torrance, CA, USA) at 450 μL/min through a 5 min gradient from 0 to 100% organic solvent B (mobile phases: A = 95% water, 5% acetonitrile, 1 mM ammonium acetate; B = 95% acetonitrile, 5% water, 1 mM ammonium acetate) in negative ion mode. Solvent gradient: 0–0.5 min 0% B, 0.5–1.1 min 0–100% B, 1.1–2.75 min hold at 100% B, 2.75–3 min 100–0% B, 3–5 min hold at 0% B. Injections were then repeated for positive ion mode at 450 μL/min through a 5 min gradient from 5 to 95% organic solvent B (mobile phases: A = water, 0.1% formic acid; B = acetonitrile, 0.1% formic acid) in positive ion mode. Solvent gradient: 0–0.5 min 5% B, 0.5–1.1 min 5–95% B, 1.1–2.75 min hold at 95% B, 2.75–3 min 95–5% B, 3–5 min hold at 5% B.

Eluant was introduced to the mass spectrometer (Thermo Q Exactive) using electrospray ionization. For both negative and positive polarities, signals were recorded at a resolution of 70,000 over a scan range of 65–900 m/z. The maximum injection time was 200 ms, microscans 2, automatic gain control (AGC) 3 × 10⁶ ions, source voltage 4.0 kV, capillary temperature 320 C, and sheath gas 45, auxiliary gas 15, and sweep gas 0 (all nitrogen)[79,80]. Resulting.raw files were converted to.mzXML format using RawConverter. Metabolites were assigned and peak areas integrated using Maven (Princeton University), in conjunction with the KEGG database and an in-house standard library of >600 compounds.

Extended methods are provided along with the submission of raw data to the public repository Metabolomics Workbench, under the Project ID number: ST002714.

## Electrophysiology

Male mice (P15 or P30) were anesthetized with a ketamine-xylazine mix (150 mg/kg) and sacrificed by decapitation. Coronal sections containing the hippocampus (250 μm thickness) were prepared and transferred for 1 h in artificial cerebrospinal fluid (ACSF: 124 mM sodium chloride; 26.2 mM sodium bicarbonate; 11 mM glucose; 2.5 mM potassium chloride; 2.5 mM calcium chloride; 1.3 mM magnesium chloride, 1 mM monosodium phosphate). Pyramidal neurons of the hippocampal CA1 region were patched using borosilicate glass pipettes (3–4 MΩ; Phymep) under an Olympus-BX51 microscope (Olympus). During recordings, slices were immersed in ACSF at 32 °C and continuously superfused at a flow rate of 2.5 ml/min. Signal was

amplified, filtered at 5 kHz and digitized at 10 kHz (MultiClamp 200B; Molecular Devices). Data were acquired using Igor Pro with NIDAQ tools (WaveMetrics). Access resistance was continuously monitored with a −4 mV step delivered at 0.1 Hz. All recordings were made in voltage-clamp configuration at −60 mV, in picrotoxin-containing ACSF (100 μM, GABAAR antagonist, HelloBio). Spontaneous excitatory postsynaptic currents were manually analyzed offline using Minianalysis (Synaptosoft, USA).

For the measurement of the AMPA/GABA ratio, all recordings were made in voltage-clamp configuration and in APV-containing ACSF (100 μM) to block NMDA receptors. To compute AMPA/GABA ratios, excitatory currents were recorded at −65 mV, while inhibitory currents were recorded at +5 mV by stimulating Schaffer collaterals with an extracellular glass pipette.

## Seizure susceptibility assessment upon kainic acid administration

Mice (P15) were injected intraperitoneally with 2 mg/kg kainic acid (KA, cat no HB0355, HelloBio) in saline and their behavior was video-ecorded and blindly scored for 2 h according to the modified Racine scale. Data were analyzed by plotting the highest behavioral score assigned to each animal over 30 min intervals, for the 2 h of observation.

## Acute microglia isolation

Mice (P15) were anesthetized with sodium pentobarbital (150 mg/kg in saline) and perfused with ice-cold PBS. The brain tissue (without olfactory bulb and cerebellum) was minced and transferred in gentleMACS C tubes (cat no 130-093-237, Miltenyi). Enzymatic dissociation was performed using the Adult Brain Dissociation Kit (cat no 130-092-628, Miltenyi), following manufacturer's instructions. After enzymatic digestion, the sample was passed through a pre-wet 70μm-filter. Cells were pelleted at 300 × g for 10 minutes at RT and myelin removal was performed as described in Mattei et al. 2020. Magnetic-based cell sorting (MACS) was performed using Cd11b-binding magnetic beads and LS columns (cat no 130-093-634 and 130-042-401, Miltenyi), following manufacturer's guidelines. Eluted Cd11b+ cells were pelleted, and either washed and lysed for downstream biochemical analysis, or alternatively resuspended in DMEM low glucose (cat no 31885049, Gibco, supplemented with 10% FBS and 1% PS) and plated in poly-D-lysine-coated 96-well plates.

## RNA extraction, cDNA synthesis and quantitative RT-PCR

Total RNA was isolated using TRI Reagent (cat no AM9738, Thermo-Fisher Scientific), following manufacturer's instructions. RNA was resuspended in UltraPure DNAse-, RNAse-free Water and DNAse treatment was performed using the DNAse I Kit (cat no EN0521, ThermoFisher Scientific). Gene expression analysis was performed via reverse transcription-quantitative PCR (RT-qPCR) and detection of amplicons was based on SYBR Green dye-emitted fluorescence (SensiFAST SYBR Lo-ROX Kit, cat no BIO94020, Bioline). cDNA was synthetized starting from 300 ng up to 1 μg total RNA, using the iScript cDNA Synthesis Kit (cat no 170-8891, Bio-Rad), and diluted with UltraPure Water to a final concentration of 1 ng/μl. Reactions were prepared in triplicate in 384-well plates (1 ng cDNA/well) and run on the ViiA 7 Real-Time PCR System (ThermoFisher Scientific). Primers pairs were designed in-house using NCBI online resources. Primers sequences are as follows. MCT1 (Slc16a1) FW: GCCGGAGTCTTTG-GATTTG; MCT1 (Slc16a1) RV: AGGCGGCCTAAAAGTGGTG; MCT2 (Slc16a7) FW: GGGCTGGGTCGTAGTCTGT; MCT2 (Slc16a7) RV: TCCAAGCGATCTGACTGGAG; MCT4 (Slc16a3) FW: CCTGGTG GTCTTTTGCATCT; MCT4 (Slc16a3) RV: TGGAGAACTTCTGAGTGCCC; Ppia FW: CAAATGCTGGACCAAACACAA; Ppia RV: GTTCATGCCTTC TTTCACCTTC; Rplp0 FW: TCGTTGGAGTGACATCGTCTT; Rplp0 RV: GATCTGCTGCATCTGCTTGG. Quantification of gene expression was

based on the deltaCt method, normalizing each gene of interest on the geometric mean of two or three housekeeping genes.

## Protein extraction and western blotting (WB)

Cells and brain tissue were lysed in ice-cold RIPA 1× (cat no 20-188, Merck), supplemented with protease inhibitors. Samples were spin at 12,000 × $g$ for 15 min at 4 °C. The supernatant was collected, and protein concentration was determined via BCA Protein Assay (cat no 23227, ThermoFisher Scientific) following manufacturer's instructions. For Western blot, 5–30 µg protein were separated on pre-casted 4-15% Mini-PROTEAN TGX Gels (cat no 4561086 and 4561084, Bio-Rad) and transferred onto nitrocellulose membranes (Trans-Blot Turbo Midi NC Transfer, cat no 170-4159, BioRad) using a semi-dry transfer protocol. Before blocking, the total amount of protein per lane was determined using REVERT Total Protein Stain Solution (cat no 926-11011, Li-Cor) and used for normalization. Blocking was performed in 3% BSA in TBST (0.02 M Tris base, 0.15 M NaCl, 0.05% Tween-20 in milliQ water, pH 7.2–7.4) for 1 h at RT and incubated overnight at 4 °C with the primary antibodies diluted in blocking buffer. Membranes were then washed 3× with TBST for 5 min at RT and incubated with secondary antibodies (IRDye, 1:15000 in blocking buffer, Li-Cor). After 3 washes with TBST, membranes were imaged with the Li-Cor Odyssey system. Uncropped blots are provided with this paper as Source Data.

## Immunofluorescence (IF)

**Cells.** cells were washed with PBS, fixed (PFA 4%, 37 °C, 20 min) and permeabilized in 0.25% Triton X-100 in PBS for 15 min at RT. Blocking was performed with 2% BSA in permeabilization buffer (30 min at RT). Primary antibodies were diluted in blocking buffer and incubated overnight at 4 °C. Cells were then washed with PBS and incubated with fluorescently labeled secondary antibodies, diluted 1:1000 in blocking buffer, for 1 h at RT. Cells were quickly washed three times with PBS and nuclei were stained for 5 min at RT with DAPI (1 µg/ml in PBS). Finally, coverslides were mounted on microscope slides using Mowiol 4-88 (cat no 81381, Sigma Aldrich). Acquisitions were taken at a Stellaris 5 confocal microscope (Leica, ×63 objective, Z-stack size 5.05µm, step size 0.3 µm) or at a Thunder Imaging System (Leica, ×40 objective, post-processing with built-in deconvolution algorithm).

**Free-floating brain slices.** fixed brains were sliced using a Vibratome VT1200S (Leica) at 60µm thickness. Slices were permeabilized in 0.5% Triton X-100 in PBS for 90 min at RT, followed by incubation with blocking buffer (2% BSA in permeabilization buffer) for 1 h at RT. Primary antibodies were diluted in blocking buffer and incubated 4 h at RT, followed by overnight incubation at 4 °C with mild agitation. Brain slices were then washed in PBS and further incubated with fluorescently labeled secondary antibodies (1:1000 in blocking buffer) for 2 h at RT. After additional washes in PBS, nuclei were stained for 10 minutes at RT with DAPI (1 µg/ml in PBS). Finally, slices were mounted on microscope slides using Mowiol 4-88 (cat no 81381, Sigma Aldrich). Acquisitions were taken at a Stellaris 5 confocal microscope (Leica), with a ×63 objective (Z-stack, step size 0.3µm, digital zoom = 1.5 for microglia morphometry and synapsin engulfment analysis; digital zoom = 4 for synaptic proteins).

## Primary antibodies

The following antibodies were used in this work, at the stated concentrations: MCT4, RRID: AB_2189333, cat. sc-50329, Santa Cruz Biotechnology (WB 1:500); IBA1, RRID: AB_839504, cat. 019-19741, FUJIFILM Wako (IF 1:1000, lot CAF6806); CD68, RRID: AB_322219, cat. MCA1957, Bio-Rad (IF 1:400, lot 155083); Synapsin1, RRID: AB_2619772, cat. 106011, Synaptic Systems (WB 1:1000, lot 1-28); Homer1, RRID: AB_887730, cat. 160003, Synaptic Systems (WB 1:1000, IF 1:200, lot 3-61); VGAT, RRID: AB_887872, cat. 131011, Synaptic Systems (WB 1:1000, lot 1-91); Gephyrin, RRID: AB_887719, cat 147111, Synaptic Systems (WB

1:1000, lot 1-25); GluA1, RRID: AB_2113602, cat. AB1504, Merck Millipore (WB 1:1000); GluA2, RRID: AB_2533058, cat. 32-0300, Thermo Fisher Scientific (WB 1:250); Synapsin1, RRID: AB_2721082, cat. 106104, Synaptic Systems (IF 1:200, lot 1-4); VGLUT1, RRID: AB_887880, cat. 135311, Synaptic Systems (WB 1:1000, lot 1-7); PSD95, RRID: AB_2092365, cat. MAB1596, Merck Millipore (WB 1:1000, IF 1:200, lot 3845700); LAMP1, RRID: AB_2134500, cat. 1D4B-c, Developmental Studies Hybridoma Bank (IF 1:100, lot 1-6-22). The MCT4 antibody was KO validated in-house, using MCT4 KO microglial samples.

## General workflow of data analysis for in vitro assays and immunofluorescence

Confocal and fluorescence acquisitions were performed using identical imaging settings across all the tested conditions and replicates for each experiment. In vitro fluorescent assays were analyzed using the Fiji software (ImageJ 1.53n)[81]. Acquisitions were taken as Z-stacks with three channels (DAPI, brightfield, and signal of interest) and the maximum projection was used for the analyses. Individual cells within the field of view were manually identified as regions of interest (ROIs) based on the cell contour on the brightfield images. The area covered of the fluorescent signal per cell was calculated upon applying the same threshold across all the tested conditions. Microglia morphometry and engulfment analyses were carried out using the Imaris software (version 9.9.1, Bitplane). 3D reconstruction was performed with the built-in "Surface" function by applying the same thresholds across all the tested conditions. All individual entire microglia cells per field of view were individually reconstructed and analyzed. For the analysis of synapsin proteins puncta and synapsin-Homer1 pairing, signal was first reconstructed using the Imaris "Spot" plugin. Colocalization analysis was performed using the colocalization spot Matlab script taking 0.3µm as the longest distance between spots.

## Dendritic spines density quantification (Golgi-cox staining)

To determine the density of dendritic spines, brain tissue was processed for Golgi-cox staining followed by confocal imaging of apical dendrites of CA1 pyramidal neurons. Specifically, P15 mice were terminally anesthetized with sodium pentobarbital (150 mg/kg in saline solution) and perfused with ice-cold PBS (10 ml total volume, over 3 minutes). The brain was isolated and processed with the FD Rapid GolgiStain Kit (cat no PK401A, FD NeuroTechnologies), following manufacturer's instruction. After two weeks impregnation in solution A + B, followed by 3 days incubation in solution C, brains were submerged in solution C and sliced with a vibratome (VT1200S, slice thickness 200 µm). Slices were mounted on gelatin-coated microscope slides and let completely dry overnight. The day after, brains were stained following manufacturer's instructions, and coverslides where mounted with Permount Mounting Medium. Acquisitions were taken at a Leica Stellaris 5 Confocal Microscope, at ×63 magnification, with a digital zoom of 2.5. Images were blinded prior to analysis for unbiased manual counting of spines by the experimenter. Only secondary and tertiary dendrites of at least 10 µm length were considered, and the first 5 µm of the dendrite from the dendrites junction was systematically excluded.

## Behavior

Behavioral characterization was performed on both males and females, at the age of 7 months. Mice were handled by the experimenter for 5 consecutive days prior to the beginning of the test battery. Mice were habituated to the room for at least 1 h before experiments. In between test sessions, the apparata were cleaned with water, followed by ethanol 70%. Where appropriate, the position of the mouse in the arena was automatically tracked from the top using the Any-maze software (version 4.99z). *Open field (OF:)* Locomotor activity and anxiety-like behavior of adult mice were evaluated in an open field arena. Light intensity was set at 10–15 lux. At the beginning of the test,

the mouse was placed in one of the corners of the apparatus (sides 45 cm, height 40 cm), facing the walls. After the mouse was released by the experimenter, it was allowed to freely explore the arena for 20 min. *Elevated plus maze (EPM):* Anxiety-like behavior was also assessed by means of the elevated plus maze test. Light intensity was set at 10–15 lux. At the beginning of the test, the mouse was placed in the center of the maze (height from the floor 60 cm, arm length 30 cm, arm width 5 cm, height 15 cm), facing one of the open arms. After being released, the mouse was allowed to freely explore the arena for 5 min. Mice were considered as "entered" in one arm only when all four paws stepped into it. *Grooming assessment:* Grooming assessment was previously described[40], with minor modifications. Light intensity was set at 40–50 lux. The mouse was placed in a transparent plastic cage (25 cm × 15 cm × 10 cm) on a thin layer of fresh bedding (1 cm). The test was recorded from the side for offline blind scoring of grooming using the BORIS software (version 8.20.4)[82]. The top of the cage was closed with a metal grid and all sides of the cage, except for the one facing the camera, were covered externally with white paper. After habituating the mouse to the new cage for 5 min, spontaneous grooming events were recorded for 10 min. To elicit stimulus-induced grooming behavior, few drops of water were sprayed on the snout of the mouse, and behavior was scored for 1 min. *Rotarod***:** Motor performance was assessed using an accelerating rotarod (Ugo Basile). Light intensity was set at 40–50 lux. Mice underwent a total of 3 trials for 2 consecutive days, with an intertrial period of 5 min. Each trial consisted in maximum 5 min total duration, with a gradual acceleration from 4 rpm to 40 rpm. *Y maze:* Working memory was assessed by means of Y maze test. Light intensity was set at 10–15 lux. At the time of testing, the mouse was placed in the center of a Y-shaped arena (arm length 35 cm, arm width 6 cm, height 16 cm) and allowed to freely explore all the arms for 3 min. The number of correct alternations between the three arms of the maze were scored blindly by the experimenter. Mice were considered as "entered" in one arm only when all four paws stepped into it. *Barnes maze test:* The Barnes maze consisted of a circular surface, with diameter 100 cm, with 20 holes (each with diameter 5 cm, distanced from the border by 4 cm). The distance of the platform from the floor was 65 cm. The test consisted in a total of 5 days, 4 trials per day. Bright light (100 lux) was shined directly onto the maze. For each trial, the mouse was first placed in the middle of the arena, covered with a dark plastic box, and was then released and allowed to explore the arena for maximum 3 min. If the mouse did not enter in the target hole at the end of the trial, it was gently guided into the target hole. On the last day, the target hole was removed, and the mouse was allowed to explore the arena for a fixed time of 90 sec.

### Statistics and reproducibility

Data representation, statistical testing, and outliers' identification were performed using GraphPad Prism (9.3.1). Results are represented as mean ± standard error of the mean. Two-tailed *t* test, one- or two-way analysis of variance were used as appropriate. Exact *p* values and biological replicates (*N*) are reported in Figure legends.

### Reporting summary

Further information on research design is available in the Nature Portfolio Reporting Summary linked to this article.

## Data availability

The transcriptomics data (bulk RNA-seq of adult microglia cells; Pinto et al.[26]), can be downloaded from the ArrayExpress database, accession number E-MEXP-3347; freely available to the following link: https://www.ebi.ac.uk/biostudies/arrayexpress/studies/E-MEXP-3347. The single-cell RNA-seq data of adult hippocampal cell populations were generated for a previous study (Mattei et al.[27]) and are accessible through GEO Series accession number GSE143796. Gene count matrices are available at https://github.com/bihealth/SC-RNA-Seq-

37vs4. The gene lists used to calculate module scores associated with cellular metabolic pathways have been downloaded from the Kyoto Encyclopedia of Genes and Genomes (KEGG database: https://www.genome.jp/kegg/). No restrictions will be placed on materials involved in this study. All data supporting the findings of this study are available within the article and its supplementary information file, provided in the Source Data file. The raw data of the metabolomics performed on hippocampal homogenates that have been generated in this study have been deposited in the public repository Metabolomics Workbench, under the Project ID number: ST002714, accessible through the following link: https://www.metabolomicsworkbench.org/data/DRCCMetadata.php?Mode=Study&StudyID=ST002714&StudyType=MS&ResultType=1. Source data are provided as a Source Data file. Source data are provided with this paper.

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

## Acknowledgements

This work was supported by grants from the Swiss National Science Foundation (SNSF 310030_197940), the Dementia Research Switzerland – Synapsis Foundation, an ERC StGrant (REMIND 804949), and funding from UNIL to RCP; Swiss National Science Foundation (SNSF 310030_212193) to MM; French Agence Nationale de la Recherche (BrainFuel ANR-21-CE44-0023-01) to LP; M.V. acknowledges funds from an ERC Consolidator Grant (DYNAFLUORS, 771443). The authors would like to thank Dr. Hector Gallart-Ayala and Dr. Julijana Ivanisevic at the Metabolomics Unit of the University of Lausanne for their technical support.

## Author contributions

K.M. designed, performed, analyzed, and interpreted most experiments and co-edited the manuscript. K.G. and A.B. performed and analyzed experiments. A.I. performed bioinformatic analyses. A.L. and A.T. performed electrophysiology experiments. A.d'A. supervised and analyzed metabolomics experiments. S.B. and M.V. provided reagents and feedback with SCOTfluor experiments. D.B., M.V., L.P., and M.M. supervised experiments. R.C.P. developed the concept, designed, performed, analyzed, and interpreted experiments, and wrote the manuscript. All authors read the manuscript and provided comments.

## Competing interests

Sam Benson and Marc Vendrell declare the following competing interests: the SCOTfluor-based fluorescent reagent (SCOTfluor510 lactate) is covered by a patent and it is commercialized by the company Tocris Bioscience. Patent applicant: The University of Edinburgh; Name of the inventor(s): Sam Benson and Marc Vendrell. Application number: WO/2020/187919. Status: Granted patent. All other authors declare no competing interests.
