## [Peer Review File · Nature Communications]

Loss of microglial MCT4 leads to defective synaptic pruning and anxiety-like behavior in miceREVIEWER COMMENTS

Reviewer #1 (Remarks to the Author):

General comments.

Monsorno et al. examined the relationship between lactate metabolism and lysosomal function in microglia. Microglial metabolism has received a great deal of attention in recent neuroscience research, and this study would bring meaningful information to the field because the authors conducted a wide range of experiments showing how the inhibition of lactate intake by MCT4 affects microglial phagocytosis and neuronal circuit remodeling. In addition to appreciating the well-designed in vitro experiments, this reviewer agrees with the authors' claim that this is the first study that aims at disrupting lactate transport in microglia in vivo. Below, I have several comments that might further strengthen the conclusions of the manuscript.

Specific comments.

1. The abbreviation LDHB firstly appeared in Page 2, Line 21. A brief explanation of LDHB as well as its formal name is necessary.

2. The authors speculate that H⁺, a byproduct of lactate metabolism, is transported to microglial lysosomes through v-ATPase, and enhances the lysosome-dependent degradation via lysosomal acidification. This hypothesis may be easily examined by using bafilomycin, a v-ATPase inhibitor.

3. The number of synapses was assessed by Western blots (Figure 4), but it would be better to perform a colocalization assay of pre- and post-synaptic markers by immunohistochemistry for more appropriate evaluation of the density of functional synapses.

4. Why is the amplitude of sEPSC increased in MCT4 cKO mice despite the increase in VGAT protein? This discrepancy may be explained by quantifying excitatory presynaptic specific protein and calculating the synaptic E/I ratio compared to the amount of inhibitory presynaptic specific protein (VGAT).

5. It would be better to plot the sEPSC data for P15 and P30 mice separately, because synaptic engulfment by microglia is suggested to be ongoing at P15 (Fig. 5).

6. Inhibition of lactate intake by MCT4 cKO would lead to the elevated extracellular concentration of lactate. In that case, it is possible that lactate itself contributes to the increased excitation of pyramidal neurons. These points can be discussed.

7. The authors investigated the effects of microglial lysosomal dysfunction on neuronal circuit remodeling and hippocampus-related behaviors. In addition to phagocytosis, the authors can discuss the possibility that lysosomal dysfunction may also affect microglial functions such as surveillance and inflammatory responses.

Reviewer #2 (Remarks to the Author):

- What are the noteworthy results? ^[1]_[SEP] The authors have used lactate- and glucose based media to test microglial cells, and mouse KOs, with respect to biochemical, morphological and behavior differences.

^[1]_[SEP] - Will the work be of significance to the field and related fields?

The reviewer fails to find significance of these studies in either of the above mentioned fields. (a) physiological tests are lacking that would impose higher, or lower, lactate concentrations in the presence of higher, or lower, glucose concentrations in both cells or mice. Presence/absence tests of nutrients have no relevance to either mice or humans as such conditions do not exist. Concentrations at 25 mM or 12.5 mM are too high to be physiologically relevant.

(b) The hypothesis tested by the authors is false or irrelevant. At best, the study has confirmatory character. The authors posited that "Lactate is a metabolic byproduct of glycolysis [...]" which is, of course, wrong. Lactate is the main product of glycolysis and it is imported by all cells. It is one of the three dominant carbon sources in mammalian blood (alanine, lactate, glucose) and in mM concentrations at any physiological condition. The authors themselves cite multiple papers as evidence of the general importance of lactate, including in the brain. The idea that all cells except microglia would import lactate would have been strange, indeed. The proof that microglia, just like all other cells, import lactate is therefore only of confirmatory value.

(c) The use of genetic knockouts in mice has limited value, as generally understood. More pointedly, if one would not know the function of a gene, a KO for such orphan genes might give clues. However, the function of MCT4 as monocarboxylate transporter was already well established with 12,500 hits in Google Scholar, and it is known to be closely associated with glycolytic activity. While genetic variants of MCT4 in human populations are known, no functional knockout in humans has been reported. It is therefore unlikely that behavioral or microglia-morphological changes in mouse KOs have any significance for human diseases. Again, physiological tests of relevance for MCT4 activity (e.g. by selective inhibitors) to mimic the effects of 10%, 30% or 50% reduced activity under high lactate and low glucose (e.g. to study developmental differences under starvation or malnutrition) might have been interesting, but were not conducted. Gene KOs do not shed light here, especially not when targeting central metabolism: central metabolism is central for cells and organs. Disabling specific genes will cause multiple pleiotropic effects (e.g. on cell morphology or behavior) which are impossible to disentangle. If one wanted to show cause-effect dependencies for lactate use (e.g. through lysosomal acidity), one would need a range of additional tests, from isotope labeling to partial or full inactivation of alternate transporters, to alternate ways to alter lysosomal pH.

How does it compare to the established literature? If the work is not original, please provide relevant references.

The reviewer was astonished to see that the following papers by Rabinowitz et al. were not cited by the authors, as the reviewer thought this was standard knowledge:

- Rabinowitz JD, Enerbäck S. Lactate: the ugly duckling of energy metabolism. *Nature Metabolism*. 2020 Jul;2(7):566-71
- Hui S, Ghergurovich JM, Morscher RJ, Jang C, Teng X, Lu W, Esparza LA, Reya T, Zhan L, Yanxiang Guo J, White E, Rabinowitz JD et al.. Glucose feeds the TCA cycle via circulating lactate. *Nature*. 2017 Nov 2;551(7678):115-8
- Many other physiological papers over the past years have studied the relevance of lactate in energy metabolism.

[SEP] Does the work support the conclusions and claims, or is additional evidence needed?

- The use of metabolomics does not support the claim or add further information. A range of figures do not further the claims by the authors, such as Suppl. Figure S4, but also Figure 5 f-k.

[SEP] Are there any flaws in the data analysis, interpretation and conclusions? - Do these prohibit publication or require revision?

- Statistical tests were not corrected for false discovery rates. The number of male and female mice were uneven for both WT and KO lines, without justification. Potential effect sizes were too small to derive any solid statistical evidence with small numbers of mice as tested here.

[SEP] Is the methodology sound? Does the work meet the expected standards in your field?

- The use of metabolomics lists unsubstantiated claims on compound identification (in the supplement data) without evidence for confidence. In-house libraries can, per definition, not be used by the scientific community and cannot serve as evidence of truth. It is unclear if the authors use accurate mass as sole criterion for (false) identifications, or if MS/MS spectra and validated retention times were used.
- Re-use of transcript data from Pinto et. al. 2012 (Suppl. Figure S1) at N=3 is unnecessary, unclear, and scientifically and statistically unsound (with three replicates and no FDR, and the known fact that most enzymatic genes have no clear correlation between transcript abundance and enzymatic activity).

SEP - Is there enough detail provided in the methods for the work to be reproduced?

- No, see above.

Reviewer #3 (Remarks to the Author):

Microglia lactate metabolism has recently become an intense area of research to modulate disease states. In this manuscript the authors aim to study the role of monocarboxylate transporter MCT4 in microglia during early development. They use in vivo and in vitro assays to conclude that microglia depletion of MCT4 in vivo leads to impaired synaptic pruning, increased excitation in hippocampal neurons enhanced vulnerability to seizure and anxiety like behavior phenotype. The studies are new in that they start to uncover the role of microglial metabolic usage of lactate in a systematic way in male and female mice, however there are many overstated conclusions that are not supported by the results presented here. The other major caveat that significantly reduces the enthusiasm for this study is that all the biochemical and histological analyses were done in early post-natal mice p15, while behavior was run in 7-8 months old mice.

It is unclear when the brain tissues were harvested for the metabolomics, ephys, engulfment assay or when the seizure susceptibility assay was run. It is essential to know the age of each assay to properly interpret the results reported here particularly because microglia undergo a profound sex dimorphism precisely at these postnatal time points.

The use of the term "synaptic pruning" is not accurate, and it is misleading particularly in the title. They did not show any structural synaptic changes in vivo, except increase of 3 synaptic proteins. Image-based quantification of spine density, or overlapped pre-post synaptic markers are needed to support this claim. Fig 4e is not convincing in that there are clearly different distribution in the cKO mice.

The cellular mechanism the authors propose is loss of MCT4-dependent lactate transportation, which is coupled with acidification of lysosomes (Fig 2 g - j), and in term deficits of lysosomal degradation of engulfed synaptosomes (Fig 3 and 4). However, in vitro phagocytosis experiments lack direct evidence for this mechanism. LysoTracker staining and lactate-free culture condition should be used in Fig 3 a - d, to prove that lactate dependent lysosomal acidification is associated with impaired lysosome degradation.

They measured 6 pre and post synaptic markers and found changes only in 3 how do they reconcile this?

Because microglia are so flexible in their metabolic programs, and they show certain metabolite changes in HPC lysates, it would be interesting to study mitochondrial changes in MCT4 cKO mice and cells.

They included both male and female data and found some sex-dependent behavioral changes but not on the cellular level this was not discussed.

The authors need a discussion/hypothesis on why molecular results are not sex dimorphic while the behavior is.

Sex differences in anxiety-like behaviors need to be articulated and discussed appropriately. They have different results in the elevated plus maze that are not discussed.

To this extent they look in CA1 area of the hippocampus known to be critical for spatial learning and memory, yet they do not report any deficits in the spatial tasks. How do they reconcile this? Behavior is run in the day cycle (not the rodent active cycle) and this could be a caveat to the results especially anxiety measures.

Point-by-point rebuttal

Monsorno et al.

Loss of microglial MCT4 leads to defective synaptic pruning and anxiety-like behavior in mice”.

Reviewer #1 (Remarks to the Author):

General comments.

Monsorno et al. examined the relationship between lactate metabolism and lysosomal function in microglia. Microglial metabolism has received a great deal of attention in recent neuroscience research, and this study would bring meaningful information to the field because the authors conducted a wide range of experiments showing how the inhibition of lactate intake by MCT4 affects microglial phagocytosis and neuronal circuit remodeling. In addition to appreciating the well-designed in vitro experiments, this reviewer agrees with the authors’ claim that this is the first study that aims at disrupting lactate transport in microglia in vivo. Below, I have several comments that might further strengthen the conclusions of the manuscript.

We thank the reviewer for the overall positive comments, and for the relevant and constructive suggestions.

Specific comments:

1. The abbreviation LDHB firstly appeared in Page 2, Line 21. A brief explanation of LDHB as well as its formal name is necessary.

We fully agree with the reviewer, and we have now included a new sentence to introduce the lactate dehydrogenase (LDH) complex, and better explain LDH subunit nomenclature, including for LDHB and its alternative names (pag.2, lines 21-25).

2. The authors speculate that H⁺, a byproduct of lactate metabolism, is transported to microglial lysosomes through v-ATPase, and enhances the lysosome-dependent degradation via lysosomal acidification. This hypothesis may be easily examined by using bafilomycin, a v-ATPase inhibitor.

We thank the reviewer for raising this important point. By repeating the experiments in the presence of bafilomycin, we have now confirmed that the effects on lysosomal acidification are dependent on V-ATPase activity. New findings are reported on pag.5, lines 25-29, and on Supplementary Figure 3g,h., and discussed on pag.10, lines 26-28.

We have further confirmed that lysosomal acidification is induced specifically by lactate, showing that pyruvate treatment does not elicit a similar effect (Figure 2k,l and pag.5, lines 21-24).

In addition, we have performed dose-dependent experiments, showing that already at 3mM concentration exogenous lactate is sufficient to promote lysosomal acidification in microglia (pag.5, lines 19-21 and Supplementary Figure 3e,f). The physiological relevance of these findings is also discussed at pag.11, lines 1-5.

3. The number of synapses was assessed by Western blots (Figure 4), but it would be better to perform a colocalization assay of pre- and post-synaptic markers by immunohistochemistry for more appropriate evaluation of the density of functional synapses.

This is a very important point, and we fully agree with the reviewer that immunofluorescence approaches are required to assess the density of functional synapses.

In this revised manuscript, we confirm by IF that synapsin levels are significantly increased in the hippocampus of cKO mice (pag.7, lines 21-22, Figure 4c,d). Furthermore, as suggested by the reviewer, we have quantified the colocalization between synapsin and the post-synaptic marker Homer1,

Faculté de biologie et de médecine

Département des sciences biomédicales

|||||

showing that the density of functional synapses is increased in cKO mice as compared to control (pag.7, lines 22-24, Figure 4c,e).

We have also performed Golgi-cox staining in the CA1 hippocampus, to quantify dendritic spine density, and found no difference between control and cKO (pag.7, lines 25-27, Figure 4f,g). These data are consistent with no changes in the levels of structural post-synaptic markers, PSD95 and Homer (as shown in Figure 4a,b and Supplementary Figure 4a,b).

4. Why is the amplitude of sEPSC increased in MCT4 cKO mice despite the increase in VGAT protein? This discrepancy may be explained by quantifying excitatory presynaptic specific protein and calculating the synaptic E/I ratio compared to the amount of inhibitory presynaptic specific protein (VGAT).

Following the suggestion of this reviewer, we have performed additional electrophysiological experiments, to quantify the relative contribution of AMPA- and GABA-mediated transmission. We have performed whole-cell patch-clamp recordings in CA1 pyramidal neurons from P15 mice, recording AMPA and GABA_A receptors-mediated currents, following electrical stimulation of the Schaffer collateral. Consistent with an increased excitatory neurotransmission, these experiments show that synapses from cKO present a higher AMPA/GABA ratio (Pag.8, lines 14-17, Figure 5d,e).

5. It would be better to plot the sEPSC data for P15 and P30 mice separately, because synaptic engulfment by microglia is suggested to be ongoing at P15 (Fig. 5).

The sEPSCs amplitude and frequency recorded at P15 and P30 are highly comparable, indicating no age-dependent effect. In the revised figure we have now made explicit the age of the mice in the plot for completeness: a code has been introduced in Figure 5c, with a detailed explanation in the Figure legend.

6. Inhibition of lactate intake by MCT4 cKO would lead to the elevated extracellular concentration of lactate. In that case, it is possible that lactate itself contributes to the increased excitation of pyramidal neurons. These points can be discussed.

This is an interesting possibility, which clearly deserves some discussion. We thank the reviewer for the suggestion. We have now included a sentence to discuss this possibility on pag.12, lines 13-16.

7. The authors investigated the effects of microglial lysosomal dysfunction on neuronal circuit remodeling and hippocampus-related behaviors. In addition to phagocytosis, the authors can discuss the possibility that lysosomal dysfunction may also affect microglial functions such as surveillance and inflammatory responses.

This is indeed an extremely important point, that we have now better explained in our discussion. We completely agree with this reviewer that microglia heavily rely on their lysosomal capacity. Therefore, lysosomal dysfunction is likely to affect other key microglial activities. This is now more clearly written on pag. 12, lines 9-12.

Reviewer #2 (Remarks to the Author):

- What are the noteworthy results? The authors have used lactate- and glucose based media to test microglial cells, and mouse KOs, with respect to biochemical, morphological and behavior differences.

- Will the work be of significance to the field and related fields?

The reviewer fails to find significance of these studies in either of the above mentioned fields.

(a) physiological tests are lacking that would impose higher, or lower, lactate concentrations in the presence of higher, or lower, glucose concentrations in both cells or mice. Presence/absence tests of nutrients have no relevance to either mice or humans as such conditions do not exist. Concentrations at 25 mM or 12.5 mM are too high to be physiologically relevant.

We agree with the reviewer that no experiments have been performed here to correlate lactate and glucose concentrations. This is out of the scope of the study.

(b) The hypothesis tested by the authors is false or irrelevant. At best, the study has confirmatory character.

We respectfully disagree with this reviewer, as we are not aware of any study that assesses the effects of lactate on microglial lysosomal acidification. Furthermore, to the best of our knowledge, no studies exist that investigate the effects of MCT4 depletion in microglia in vivo.

The authors posited that “Lactate is a metabolic byproduct of glycolysis [...]” which is, of course, wrong. Lactate is the main product of glycolysis and it is imported by all cells. It is one of the three dominant carbon sources in mammalian blood (alanine, lactate, glucose) and in mM concentrations at any physiological condition. The authors themselves cite multiple papers as evidence of the general importance of lactate, including in the brain. The idea that all cells except microglia would import lactate would have been strange, indeed. The proof that microglia, just like all other cells, import lactate is therefore only of confirmatory value.

We absolutely agree with this reviewer that lactate is indeed a key metabolite.

At pag.2, lines 28-29, we write that “lactate is not merely a waste product of glycolysis, but rather a key metabolite for inter-cellular communication”, fully agreeing with the perspective of this reviewer. We have now corrected the word “byproduct” with “product” (pag.2 line 16).

We agree that the fact that microglia could import lactate is not surprising. Nevertheless, studies that assess the functional consequences of lactate import into microglia are lacking. With this study, we aimed indeed to shed light on the functional modulation that is downstream of lactate catabolism, which can very much depend on the specific cell type of investigation.

(c) The use of genetic knockouts in mice has limited value, as generally understood. More pointedly, if one would not know the function of a gene, a KO for such orphan genes might give clues. However, the function of MCT4 as monocarboxylate transporter was already well established with 12,500 hits in Google Scholar, and it is known to be closely associated with glycolytic activity. While genetic variants of MCT4 in human populations are known, no functional knockout in humans has been reported. It is therefore unlikely that behavioral or microglia-morphological changes in mouse KOs have any significance for human diseases.

We agree with the reviewer that the role of MCT4 as a lactate transporter is well established in the literature. In this regard, we would also like to mention that, nevertheless, this area of investigation is still very active and more recent studies have considerably challenged the traditional beliefs concerning the biochemical properties of MCT4 (see Contreras-Baeza et al. 2019), although this was not the scope of our research.

Faculté de biologie et de médecine
Département des sciences biomédicales

More importantly, the functional consequences of lactate transport disruption have been mainly studied in neoplastic cells, which cannot be considered comparable to other cell types and to the physiological regulation of their function. Of note, a Pubmed search for the keyword “MCT4” returns 866 studies, while “MCT4 + microglia” only 3 hits.

Given the increasing evidence linking intracellular metabolism to immune-related cellular function, more specific studies are needed to fill this gap of knowledge in physiologically relevant cell types.

Conditional knockout mouse lines, as the one used in this study, are powerful tools to dissect the cellular contribution of given genes to a complex phenotype.

Finally, we do not make any overstatement about human disease. Nevertheless, snRNA-seq from adult human brains (Habib et al. 2017; Dataset available on singlecell.broadinstitute.com) clearly shows that MCT4 in the human brain is enriched in microglia (Cx3cr1+ and P2RY12+ cells), while MCT1 and MCT2 are not, providing ground for MCT4 implication in human microglial lactate metabolism. We attach here the data:

MCT4 expression in human microglia

Again, physiological tests of relevance for MCT4 activity (e.g. by selective inhibitors) to mimic the effects of 10%, 30% or 50% reduced activity under high lactate and low glucose (e.g. to study developmental differences under starvation or malnutrition) might have been interesting, but were not conducted. Gene KO's do not shed light here, especially not when targeting central metabolism: central metabolism is central for cells and organs. Disabling specific genes will cause multiple pleiotropic effects (e.g. on cell morphology or behavior) which are impossible to disentangle. If one wanted to show cause-effect dependencies for lactate use (e.g. through lysosomal acidity), one would need a range of additional tests, from isotope labeling to partial or full inactivation of alternate transporters, to alternate ways to alter lysosomal pH.

We thank the reviewer for the suggestion. However, inhibitors selective for MCT4 are currently lacking. This was indeed one of the reasons to base this study on the use of conditional KO mouse

models, that allow us to selectively disrupt MCT4 avoiding the risk of off-target effects, which are very common for MCTs inhibitors and not completely characterized.

We have also included additional experiments aimed at titrating the concentration of extracellular lactate (Pag.5, lines 19-21, Supp. Figures 3e,f). These data show that the effect of lactate on lysosomal acidification is significant starting from 3mM and it is stably present at increasing concentrations, including 25mM.

How does it compare to the established literature? If the work is not original, please provide relevant references.

The reviewer was astonished to see that the following papers by Rabinowitz et al. were not cited by the authors, as the reviewer thought this was standard knowledge:

- Rabinowitz JD, Enerbäck S. Lactate: the ugly duckling of energy metabolism. *Nature Metabolism*. 2020 Jul;2(7):566-71
- Hui S, Ghergurovich JM, Morscher RJ, Jang C, Teng X, Lu W, Esparza LA, Reya T, Zhan L, Yanxiang Guo J, White E, Rabinowitz JD et al.. Glucose feeds the TCA cycle via circulating lactate. *Nature*. 2017 Nov 2;551(7678):115-8
- Many other physiological papers over the past years have studied the relevance of lactate in energy metabolism.

We thank the reviewer for the comment. We indeed agree that many physiological papers have studied the relevance of lactate in energy metabolism. Therefore, we have found more appropriate to cite studies relevant to the organ of investigation (i.e. brain) as well as focus on studies related to immune cells, that resemble more closely the physiology of microglial cells. We have now included in the discussion the citation of the review *"Rabinowitz JD, Enerbäck S. Lactate: the ugly duckling of energy metabolism. Nature Metabolism. 2020 Jul;2(7):566-71 and the article Hui et al., 2017.*

- Does the work support the conclusions and claims, or is additional evidence needed?
- The use of metabolomics does not support the claim or add further information. A range of figures do not further the claims by the authors, such as Suppl.Figure S4, but also Figure 5 f-k.
- Are there any flaws in the data analysis, interpretation and conclusions? - Do these prohibit publication or require revision?

- Statistical tests were not corrected for false discovery rates. The number of male and female mice were uneven for both WT and KO lines, without justification. Potential effect sizes were too small to derive any solid statistical evidence with small numbers of mice as tested here.

- Is the methodology sound? Does the work meet the expected standards in your field?

- The use of metabolomics lists unsubstantiated claims on compound identification (in the supplement data) without evidence for confidence. In-house libraries can, per definition, not be used by the scientific community and cannot serve as evidence of truth. It is unclear if the authors use accurate mass as sole criterion for (false) identifications, or if MS/MS spectra and validated retention times were used.

We appreciate the reviewer for bringing up this relevant point. Metabolites were identified via UHPLC-HRMS, with determination of chemical formula based on high resolution intact mass, isotopic pattern and further validation via retention times for standard compounds. While an additional degree of confidence is provided by MS/MS identification, high-throughput approaches relying on ballistic

Faculté de biologie et de médecine
Département des sciences biomédicales

|||||

gradients and/or flow-injections often rely on accurate intact mass. While the reviewer may disagree, data generated with these workflows are routinely accepted by the metabolomics community (see the thematic book on High-Throughput Metabolomics – by Springer Nature edited by one of the co-authors of this study - <https://link.springer.com/book/10.1007/978-1-4939-9236-2>), including all leading journals from the Nature publishing group (see link here). For example, similar analytical workflows have been used in other papers from the D’Alessandro lab – that generated the data presented herein – as published in 2023 in this very journal Nature Communications (e.g., Turner et al. Nature Comm 2023; accepted on May 19, 2023 – NCOMMS-22-37094B).

- Re-use of transcript data from Pinto et. al. 2012 (Suppl. Figure S1) at N=3 is unnecessary, unclear, and scientifically and statistically unsound (with three replicates and no FDR, and the known fact that most enzymatic genes have no clear correlation between transcript abundance and enzymatic activity).

We thank the reviewer for the comment. However, in this paper we did not include any statistical analyses on these gene expression database. These data were displayed only with qualitative value and therefore treated and discussed as such. We have now included these data in the Supplementary Figure 1 and we have added to Figure1 the comparison of metabolic genes expression across multiple brain cell types, re-analyzing our own dataset from scRNA-seq of the adult mouse hippocampus (PMID: 33114694). In this case, statistical comparison is provided. This additional analysis underlines the remarkable potential of microglia for a higher metabolic flexibility, at least at the transcriptional level.

- Is there enough detail provided in the methods for the work to be reproduced?

- No, see above.

Extended methods are provided along with the submission of raw data to the public repository Metabolomics Workbench, under the Project ID number: ST002714.

Reviewer #3 (Remarks to the Author):

Microglia lactate metabolism has recently become an intense area of research to modulate disease states. In this manuscript the authors aim to study the role of monocarboxylate transported MCT4 in microglia during early development. They use in vivo and in vitro assays to conclude that microglia depletion of MCT4 in vivo leads to impaired synaptic pruning, increased excitation in hippocampal neurons enhanced vulnerability to seizure and anxiety like behavior phenotype. The studies are new in that they start to uncover the role of microglial metabolic usage of lactate in a systematic way in male and female mice, however there are many overstated conclusions that are not supported by the results presented here. The other major caveat that significantly reduces the enthusiasm for this study is that all the biochemical and histological analyses were done in early post-natal mice p15, while behavior was run in 7-8 months old mice.

We thank the reviewer for the overall positive feedback and for the relevant comments and suggestions. We have now corroborated our findings, to further support the conclusions of the study. Furthermore, we have included additional biochemical and immunofluorescence analyses from the adult brains to complement the behavioral characterization done in adulthood.

It is unclear when the brain tissues were harvested for the metabolomics, ephys, engulfment assay or when the seizure susceptibility assay was run. It is essential to know the age of each assay to properly interpret the results reported here particularly because microglia undergo a profound sex dimorphism precisely at these postnatal time points.

We completely agree with the reviewer and apologize for the confusion. We now make sure that all the exact time points are properly indicated in each figure legend as well as in the main text body.

The use of the term “synaptic pruning” is not accurate, and it is misleading particularly in the title. They did not show any structural synaptic changes in vivo, except increase of 3 synaptic proteins. Image-based quantification of spine density, or overlapped pre-post synaptic markers are needed to support this claim. Fig 4e is not convincing in that there are clearly different distribution in the cKO mice.

We agree with this reviewer that structural changes of synapses are required to further support the statement that pruning is impaired in cKO mice.

We have now performed additional immunofluorescence experiments, which show increased levels of hippocampal synapsin in cKO mice at P15 (Pag.7, lines 21-22, Figure 4c,d), corroborating structural synaptic changes.

Furthermore, we have assessed co-localization between pre- and post-synaptic markers (synapsin and homer1), showing that the density of functional synapses is also increased in cKO mice (Pag.7, lines 22-24, Figure 4c,e).

We have also performed dendritic spine density analysis, by Golgi-Cox staining, showing no differences in spine density, consistent with no alterations in the levels of postsynaptic scaffold proteins such as Homer1 and PSD95 (Pag.7, lines 25-27, Figure 4f,g). These data would point toward a specific effect on pre-synaptic terminals. A paragraph has been introduced in the discussion, at pag. 11, lines 21 to 28.

Finally, we have further corroborated the data about synaptic pruning by increasing the number of biological replicates in the synapsin engulfment analysis by microglia. We have now included n=30 control cells vs. n=42 cKO cells, from N=7 mice per genotype (3 males and 4 females). The total number

and volume of engulfed synapsin in KO microglia is now significantly higher than controls (**p = 0.0001 and ***p < 0.0001, respectively). These data are reported at pag. 8, lines 1-3, Figure 4h-j.

The cellular mechanism the authors propose is loss of MCT4-dependent lactate transportation, which is coupled with acidification of lysosomes (Fig 2 g - j), and in term deficits of lysosomal degradation of engulfed synaptosomes (Fig 3 and 4). However, in vitro phagocytosis experiments lack direct evidence for this mechanism. LysoTracker staining and lactate-free culture condition should be used in Fig 3 a - d, to prove that lactate dependent lysosomal acidification is associated with impaired lysosome degradation.

The point raised by the reviewer is very important. We indeed found that in MCT4 KO primary microglia the decrease in LysoTracker signal is associated with decreased efficiency of DQ-BSA and impaired synaptosomal degradation (Figure 3a,b and Figure 3c,d). In this revised manuscript, following the suggestion of the reviewer, we also provide evidence that in lactate-free culture medium no differences between control and cKO microglia in LysoTracker signal are observed (Pag.5, lines 9-10, Supplementary Figure 2b,c), overall supporting a direct role for lactate in modulating lysosomal acidification through MCT4.

They measured 6 pre and post synaptic markers and found changes only in 3 how do they reconcile this?

Microglia-mediated synaptic refinement is a complex process, which can differentially affect pre- and post-synaptic structures, as well as subset of synapses over others, possibly depending on the timing and triggers. Moreover, the nibbling of portions of the synaptic terminals, rather than the stripping of the whole synapse, is likely to occur, therefore explaining why certain synaptic markers might be more affected than others. In our study, we had on purpose screened for a number of pre- and post-synaptic markers, to better assess the global picture, which reflects the complexity of the process. This aspect has been now discussed at pag. 11, lines 21-28.

Because microglia are so flexible in their metabolic programs, and they show certain metabolite changes in HPC lysates, it would be interesting to study mitochondrial changes in MCT4 cKO mice and cells.

We absolutely agree with this reviewer that the mitochondrial changes induced in microglia by lactate exposure, in control and cKO mice and cells, is of utmost importance. However, these studies require time and resources, and we are currently performing them for a more metabolically focused manuscript, which will be a follow-up of the present study.

They included both male and female data and found some sex-dependent behavioral changes but not on the cellular level this was not discussed.

The authors need a discussion/hypothesis on why molecular results are not sex dimorphic while the behavior is.

Sex differences in anxiety-like behaviors need to be articulated and discussed appropriately. They have different results in the elevated plus maze that are not discussed.

Thanks for raising this important aspect. We have now expanded the discussion to both highlight the sex-dimorphic phenotypes at the cellular and behavioral level, in a dedicated paragraph at pag. 13, from line 13 to 21.

To this extent they look in CA1 area of the hippocampus known to be critical for spatial learning and memory, yet they do not report any deficits in the spatial tasks. How do they reconcile this?

A growing body of literature supports the implication of the hippocampus in anxiety disorders
PMID: 25152721.

We agree with the reviewer that in the presence of CA1 hippocampal defects, one would expect to observe defective spatial learning and memory. However, the hippocampus is a highly complex structures with tight substructural regulations implicated in different behavioral modulation (Fanselow and Dong, 2010, PMID: 20152109). For instance, cortico-hippocampal circuits (Entorhinal cortex-dorsal hippocampus) have been shown to support the encoding of spatial, episodic, social, and contextual memories. On the other hand, ventral hippocampus-amygdala, seem to be more relevant for anxiety and stress (Fanselow and Dong, 2010, PMID: 20152109). We have now included a sentence in the discussion to highlight this aspect (pag.13, lines 19-21).

It would be definitely interesting to explore -at the circuit level- how MCT4 cKO influences hippocampal maturation and function, in a different study.

We have also included new data about synaptic changes in the adult hippocampus (pag.10, lines 1-4, Figure 7) and further discussed them on pag.13 (lines 21-28).

Behavior is run in the day cycle (not the rodent active cycle) and this could be a caveat to the results especially anxiety measures.

We understand the reviewer's concern, however it is important to notice that all mice (control and cKO) have been subjected to behavioral tests at the same time. Therefore, any potential effect on anxiety behavior would have equally affect both genotypes. Nevertheless, only cKO mice show an increase in anxiety-like behavior. In addition, all mice were extensively handled by the experimenter for at least five consecutive days prior to testing, in order to reduce any stress component on the actual behavioral tasks.

REVIEWERS' COMMENTS

Reviewer #1 (Remarks to the Author):

The authors have properly addressed my comments and I have no additional comments.

Reviewer #2 (Remarks to the Author):

The authors have responded tactfully to the criticism raised by the reviewer. Unfortunately, the authors could not refute the fundamental concerns the reviewer had with this manuscript with respect to physiologically unrealistic concentrations, poor compound annotation confidence lacking MS/MS data, and poor study designs with relying on KO mutants. The main criticisms remain unanswered, unfortunately

Reviewer #3 (Remarks to the Author):

The authors addressed all the comments

Rosa Chiara Paolicelli
Université de Lausanne
Department of Biomedical Sciences (DBS)
Rue du Bugnon, 7 – CH-1005 Lausanne
E-mail: rosachiara.paolicelli@unil.ch
Tel: +41 21 692 55 20

August 23rd 2023

Point-by-point rebuttal

Monsorino et al.

“Loss of microglial MCT4 leads to defective synaptic pruning and anxiety-like behavior in mice”.

Reviewer #1 (Remarks to the Author):

The authors have properly addressed my comments and I have no additional comments.
We are glad to have properly addressed all the concerns of the reviewer.

Reviewer #2 (Remarks to the Author):

The authors have responded tactfully to the criticism raised by the reviewer. Unfortunately, the authors could not refute the fundamental concerns the reviewer had with this manuscript with respect to physiologically unrealistic concentrations, poor compound annotation confidence lacking MS/MS data, and poor study designs with relying on KO mutants. The main criticisms remain unanswered, unfortunately.

We hope that titration experiments showing that extracellular lactate induces a significant increase in lysosomal acidification already at 3mM concentration will appease this reviewer’s concerns.
All the findings *in vivo* comparing MCT4 cKO vs. control mice rely exclusively on endogenous physiological concentrations of metabolites, as no exogenous treatment is applied.
As for the criticism on the use of mouse genetics to KO the MCT4 transporter in a specific cell type, we respectfully argue that no other means would serve the same scope, since selective MCT4 inhibitors -as the reviewer was suggesting using- do not currently exist, and certainly their activity *in vivo* would not be cell specific.

Regarding the mass spectrometry data, we have now included the following dedicated method section in the manuscript, reporting the detailed protocols for brain tissue processing and extraction:

“Mice (P15) were sacrificed by decapitation and the brain regions of interest were quickly dissected and snap frozen in liquid nitrogen. Specifically, frozen tissue samples were extracted at 20mg/mL using ice-cold 5:3:2 methanol:acetonitrile:water. Samples were homogenized using a bead beater for 5min, on ice. The homogenates were then vortexed 30min and spun down for 10min at 18000g at 4°C. After sample randomization, 10 µL of extracts were injected into a Thermo Vanquish UHPLC system (San Jose, CA, USA) and resolved on a Kinetex C18 column (150 × 2.1 mm, 1.7 µm, Phenomenex, Torrance, CA, USA) at 450 µL/min through a 5 min gradient from 0 to 100% organic solvent B (mobile phases: A

Faculté de biologie et de médecine
Département des sciences biomédicales

|||||

= 95% water, 5% acetonitrile, 1 mM ammonium acetate; B = 95% acetonitrile, 5% water, 1 mM ammonium acetate) in negative ion mode. Solvent gradient: 0-0.5 min 0% B, 0.5-1.1 min 0-100% B, 1.1-2.75 min hold at 100% B, 2.75-3 min 100-0% B, 3-5 min hold at 0% B. Injections were then repeated for positive ion mode at 450 μ L/min through a 5 min gradient from 5 to 95% organic solvent B (mobile phases: A = water, 0.1% formic acid; B = acetonitrile, 0.1% formic acid) in positive ion mode. Solvent gradient: 0-0.5 min 5% B, 0.5-1.1 min 5-95% B, 1.1-2.75 min hold at 95% B, 2.75-3 min 95-5% B, 3-5 min hold at 5% B.

Eluent was introduced to the mass spectrometer (Thermo Q Exactive) using electrospray ionization. For both negative and positive polarities, signals were recorded at a resolution of 70,000 over a scan range of 65-900 m/z. The maximum injection time was 200 ms, microscans 2, automatic gain control (AGC) 3×10^6 ions, source voltage 4.0 kV, capillary temperature 320 C, and sheath gas 45, auxiliary gas 15, and sweep gas 0 (all nitrogen)^{68,69}. Resulting .raw files were converted to .mzXML format using RawConverter. Metabolites were assigned and peak areas integrated using Maven (Princeton University), in conjunction with the KEGG database and an in-house standard library of >600 compounds".

We have also clearly stated that extended methods are provided along with the submission of raw data to the public repository Metabolomics Workbench, under the Project ID number: ST002714, and the hyperlink to freely access this dataset is included in the text.

Quantified data, including peak intensity and area, across the LC-MS experiments are also provided in an .xls file as a "Supplementary dataset".

As discussed in the previous rebuttal, for LC-MS data generation we rely on workflows routinely accepted by the metabolomics community, reported in the thematic book on High-Throughput Metabolomics edited by Prof. D'Alessandro, a co-author of this manuscript, by Springer Nature - <https://link.springer.com/book/10.1007/978-1-4939-9236-2>.

Furthermore, as previously discussed, similar analytical workflows have been used in other papers from the D'Alessandro lab – that generated the data presented herein – as published in 2023 in this very journal Nature Communications (e.g., Turner et al. Nature Comm 2023; accepted on May 19, 2023 – NCOMMS-22-37094B).

Reviewer #3 (Remarks to the Author):

The authors addressed all the comments.

We are glad to have properly addressed all the comments of the reviewer.